# Closing the Safety Gap: Surgical Concept Erasure in Visual Autoregressive Models

Xinhao Zhong[1*]    Yimin Zhou[2*]    Zhiqi Zhang[3]    Junhao Li[1]
Yi Sun[1]    Bin Chen[1,4†]    Shu-Tao Xia[2]    Xuan Wang[1]    Ke Xu[5]
[1]Harbin Institute of Technology, Shenzhen
[2]Tsinghua Shenzhen International Graduate School, Tsinghua University
[3]Jilin University    [4]Peng Cheng Laboratory
[5] Department of Computer Science and Technology, Tsinghua University

## Abstract

The rapid progress of visual autoregressive (VAR) models has brought new opportunities for text-to-image generation, but also heightened safety concerns. Existing concept erasure techniques, primarily designed for diffusion models, fail to generalize to VARs due to their next-scale token prediction paradigm. In this paper, we first propose a novel VAR Erasure framework **VARE** that enables stable concept erasure in VAR models by leveraging auxiliary visual tokens to reduce fine-tuning intensity. Building upon this, we introduce **S-VARE**, a novel and effective concept erasure method designed for VAR, which incorporates a filtered cross entropy loss to precisely identify and minimally adjust unsafe visual tokens, along with a preservation loss to maintain semantic fidelity, addressing the issues such as language drift and reduced diversity introduce by naïve fine-tuning. Extensive experiments demonstrate that our approach achieves surgical concept erasure while preserving generation quality, thereby closing the safety gap in autoregressive text-to-image generation by earlier methods. Our code is available at https://github.com/ndhg1213/S-VARE.

## 1 Introduction

The rapid progress of text-to-image generative models (Rombach et al., 2022; Ramesh et al., 2022; Labs, 2024; Han et al., 2025) has significantly enhanced their ability to produce high-quality outputs with strong adherence to text prompt. These advancements are driven not only by improvements in model architectures but also by the availability of large-scale training data (Schuhmann et al., 2022; Byeon et al., 2022). More recently, a new family of generative models known as visual autoregressive models (VAR) (Tian et al., 2024; Han et al., 2025) has been introduced. Unlike traditional autoregressive models that generate visual tokens in a raster-scan order (Chen et al., 2025), VAR models predict visual tokens at progressively larger scale. This hierarchical generation paradigm brings substantial improvements in both image quality and generation speed. Notably, Infinity (Han et al., 2025) has showcased the strong performance of VAR models on high-resolution text-to-image generation tasks. Despite their advantages, current text-to-image VAR models lack effective safety mechanisms and remain susceptible to generating sensitive or inappropriate images. However, such models are often capable of generating unsafe content in response to inappropriate prompts, such as NSFW (Not-Safe-For-Work) images involving pornography and violence (Jiang et al., 2023), or material that may raise copyright concerns (Qu et al., 2023). A more pressing challenge emerges when new undesirable concepts are identified after model training. Reconstructing the training dataset and retraining the model from scratch for each such case imposes an impractical computational burden. This significantly hinders the safe and scalable deployment of text-to-image generation systems in real-world applications.

---

[*]Equal Contribution.
[†]Corresponding Author.

Concept Erasure (CE), a family of emerging methods serves as a promising solution for efficiently removing undesirable concepts from generative models. These approaches achieve concept erasure by modifying model components (e.g., cross-attention modules) or Low-Rank Adaptation (LoRA) (Hu et al., 2022) through fine-tuning(Gandikota et al., 2023; Zhang et al., 2024a; Kumari et al., 2023), closed-form solutions (Gandikota et al., 2024), or neuron pruning (Chavhan et al., 2024; Sun et al., 2026). Existing methods have been well-studied in the domain of diffusion models (Rombach et al., 2022) based on U-Net (Ronneberger et al., 2015) architectures and follow-up works (Zhang et al., 2025; Gao et al., 2025) have also extended these techniques to FLUX (Labs, 2024), a transformer-based (Vaswani et al., 2017) diffusion model that employs flow matching (Lipman et al., 2022), demonstrating some degree of success. However, existing methods developed for diffusion models cannot be directly applied to VAR models, due to the different use of visual GPT-based (Hurst et al., 2024) transformer and the fact that the prediction targets are visual tokens instead of noise. This results in a clear methodological gap in concept erasure for this emerging family of generative models.

In this work, we examine the limitations of existing CE methods originally developed for diffusion models when applied to VAR framework. Current approaches align visual tokens independently at each scale using differential prompts, a procedure reminiscent of aligning diffusion outputs at individual timesteps. However, this independent alignment introduces cumulative errors across scales, often leading to severe degradation in image quality. To address this challenge, we first introduce **VARE**rasure, a framework that leverages auxiliary target tokens as additional inputs to mitigate discrepancies caused by token misalignment. Building on this framework, we further propose **S-VARE**, a method for surgical concept erasure in VAR models. Unlike prior work that formulates erasure as a regression problem by minimizing mean squared error (MSE) between predicted and reference noise in U-Nets, recent advances such as the Infinity model have demonstrated that binary spherical quantization (BSQ) (Zhao et al., 2024) can improve codebook efficiency by projecting predictions into a probability space. Inspired by this, we design a filtered cross-entropy loss $\mathcal{L}_{FCE}$ that measures semantic differences more precisely by computing bit-wise discrepancies between predicted tokens and quantized targets. Finally, to counter common side effects of naïve fine-tuning, such as language drift and reduced output diversity, we introduce a preservation loss $\mathcal{L}_{Pre}$ tailored for VARs, which aligns outputs of the pre-trained and fine-tuned models, safeguarding unrelated concepts and maintaining generative diversity. In summary, we make the following contributions:

- We are the first to systematically analysis the challenge of directly applying existing diffusion-based CE methods to VAR models, and we propose a novel fundamental **VARE** framework to address this limitation.

- By analyzing the characteristics of the VAR framework, we identify the limitations of existing erasure functions and introduce a new concept erasure method **VARE**, which consists of a filtered erasure loss $\mathcal{L}_{FCE}$ and a preservation loss $\mathcal{L}_{Pre}$.

- Extensive experimental results demonstrate that our method successfully erases 97% of sensitive concepts while causing less than 2% degradation in CLIP score, filling a critical gap in the safe and efficient deployment of text-to-image generation models.

## 2 RELATED WORKS

### 2.1 VISUAL AUTOREGRESSIVE MODELS

To unify visual generation and understanding within a single framework, autoregressive visual generation models typically first apply vector quantization (VQ) to convert image patches into discrete visual tokens (Van Den Oord et al., 2017). These models then predict the next visual token in a determined raster scan order conditioned on previously generated tokens (Yu et al., 2024; Fan et al., 2024). Building on this pipeline, numerous works (Chen et al., 2025; Deng et al., 2025; Wang et al., 2024a;b) have developed increasingly powerful architectures for image and video generation tasks. Recently, Visual Autoregressive Models (VAR) (Tian et al., 2024) introduced a novel next-scale prediction paradigm that significantly improves generation quality. Subsequent works (Li et al., 2024; Yao et al., 2024; Zhang et al., 2024b) have explored controllable generation within the VAR framework, and Infinity (Han et al., 2025) further advances this line of work by employing bit-wise quantization to enhance scalability and achieves high-quality text-to-image generation with strong instruction fidelity.

## 2.2 Concept Erasure

To mitigate safety risks in text-to-image diffusion models, such as the generation of NSFW or copyright-sensitive content, concept erasure (CE) has emerged as a more efficient and principled alternative to pre-training filtering (Rombach et al., 2022) or post-generation filtering (Rando et al., 2022). The goal is to remove the model's ability to generate images containing undesired concepts while preserving its overall generative capacity. Existing methods typically align model outputs with and without concept-conditioned prompts. FMN (Zhang et al., 2024a) minimizes attention activations associated with target concept text tokens. ESD (Gandikota et al., 2023) and CA (Kumari et al., 2023) fine-tune cross-attention modules by aligning predicted noise via MSE. UCE (Gandikota et al., 2024) solves a closed-form optimization of the text projection matrix. Subsequent works (Lu et al., 2024; Zhang et al., 2024c; Bui et al., 2024; Kim et al., 2024) leverage techniques such as LoRA and adversarial training to improve erasure precision while maintaining generation quality, and recent works (Zhang et al., 2025; Gao et al., 2025) have extended concept erasure to FLUX, a transformer-based diffusion model with flow matching. However, existing approaches are constrained to the diffusion paradigm and are not directly applicable to VAR models, which differ fundamentally in architecture and generation dynamics. This work identifies the key obstacles in adapting CE methods to VAR and introduces the first effective erasure method tailored for VAR models.

## 3 Preliminaries

In autoregressive visual generation, the image is first encoded into a latent representation following a fixed raster scan manner and then quantized into a sequence of discrete tokens $t = \{t_1, t_2, \cdots, t_N\}$. During inference, the model predicts the next visual token $t_n$ conditioned on all previously generated tokens $t_{<n} = \{t_1, t_2, \cdots, t_{n-1}\}$ and the condition $c$. The predicted probabilities of whole image can be formulated as follow:

$$p(x) = \prod_{n=1}^{N} p(t_n \mid t_{<n}, c). \tag{1}$$

VAR (Tian et al., 2024) redefines the autoregressive pipeline objective by predicting the next scale visual tokens. Given an image $x$, VAR first encodes it into continuous feature representations $f \in \mathbb{R}^{h \times w \times C}$, which are then quantized into a $K$ level residual token maps $r = \{r_1, r_2, \cdots, r_K\}$, each scale map $r_i$ contains $h_i \times w_i$ tokens $t \in \mathbb{R}^{2 \times d}$, where $d$ is the vocabulary size of the VQ-VAE. Based on this residual sequence, $f_k$ at each scale $k$ can be reconstructed as below:

$$f_k = \sum_{i=1}^{K} \text{upsample}(\text{lookup}(r_i)), \tag{2}$$

where upsample($\cdot$) refers to linear upsampling and lookup($\cdot$) refers to matching codebook. $f_k$ at each level is the cumulative sum of lower-scale features. The visual transformer predicts the residuals $r_k$ at the next scale, conditioned on the existing residual sequence $r_{<k} = \{r_1, r_2, \cdots, r_{k-1}\}$ and condition $c$. The overall generation process can be formalized as follows.

$$p(r) = \prod_{i=1}^{K} p(r_i \mid r_{<i}, c), \tag{3}$$

To address the computational overhead associated with an expanded codebook, Infinity (Han et al., 2025) replaces the original vector quantizer in VAR with a bit-wise quantizer. For each input vector $z \in \mathbb{R}^d$, BSQ (Zhao et al., 2024) is applied to obtain a binary output $q$ as defined below:

$$q = \frac{1}{\sqrt{d}} \text{sign}\left(\frac{z}{\mid z \mid}\right), \tag{4}$$

where sign($\cdot$) denotes the signum function. By transforming the prediction target into a bit-wise representation, Infinity successfully extends the VAR framework to large-scale text-to-image generation.

## 4 Method

In this section, we delve into the specifics of our method, aiming to address the limitations of directly applying diffusion-based CE methods to VAR models. The overall framework is shown in Figure 2.

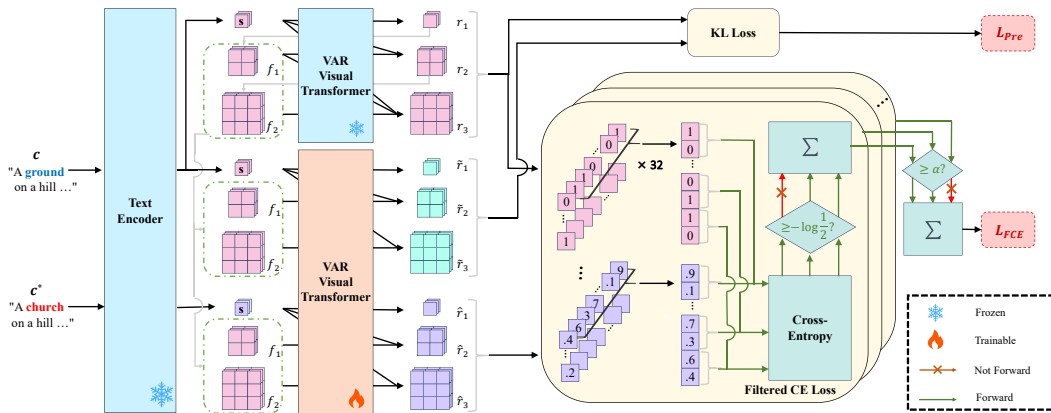

Figure 1: The framework of our method. The left part illustrates the proposed erasure framework adapted for VAR models, while the right part presents the proposed filtered cross entropy loss $\mathcal{L}_{FCE}$ and the preservation loss $\mathcal{L}_{Pre}$.

## 4.1 VAR ERASURE FRAMEWORK WITH AUXILIARY VISUAL TOKENS

Diffusion-based CE methods align the predicted noise generated under prompt $c^*$ which contains a target concept with the predicted noise generated under neutral prompts $c$ which excludes the concept. This process can be formalized as follows.

$$\mathcal{L}_{era} = \mathbb{E}_t[||\epsilon_{\theta^*}(x_t, c^*, t) - \epsilon_\theta(x_t, c, t)||_2^2], \tag{5}$$

where $x_t$ denotes the noised latent at timestep $t$, $\theta^*$ and $\theta$ represent the trainable fine-tuned model parameters and original model parameters, respectively. A straightforward approach to adapting existing CE methods to the VAR framework is to replace the predicted noise $x_t$ at each diffusion timestep in Eq. (5) with the predicted visual tokens $r_i$ at each scale, as shown below:

$$\mathcal{L}_{vanilla} = \mathbb{E}_i[||p_{\theta^*}(r_i \mid r_{<i}, c^*) - p_\theta(r_i \mid r_{<i}, c)||_2^2], \tag{6}$$

However, unlike diffusion models where denoising steps are relatively independent across timesteps, VAR generation is highly autoregressive: each token prediction depends heavily on previously predicted tokens at coarser scales. Consequently, optimization with Eq. (6) introduces errors that accumulate progressively across scales, eventually causing severe image quality collapse, as illustrated in the left column of Figure 2.

To mitigate this issue, we provide the VAR visual transformer with auxiliary visual tokens as additional inputs, which serve as references to stabilize generation. To further reduce the optimization search space, we incorporate tokens predicted by $p_\theta(r_i \mid r_{<i}, c^*)$ and $p_\theta(r_i \mid r_{<i}, c)$ during training, where $r^{ori,*}$ and $r^{ori}$ denote the tokens generated from prompts $c^*$ and $c$, respectively. As shown in the middle column of Figure 2, using $r^{ori,*}$ alleviates collapse to some extent, but large discrepancies between target and auxiliary tokens still prevent faithful generation. In contrast, when $r^{ori}$ are provided (see left of Figure 1), the model only needs to adjust cross-attention responses to account for the effect of $c^*$, leaving the overall behavior largely

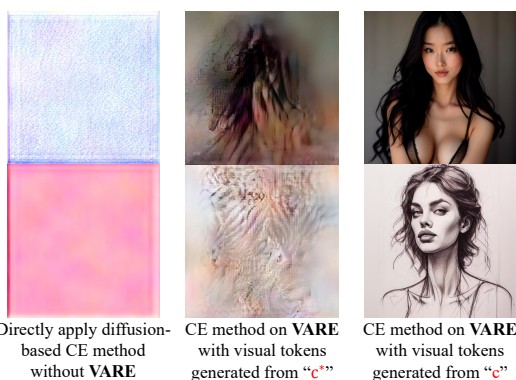

Directly apply diffusion-based CE method without **VARE** | CE method on **VARE** with visual tokens generated from "$c^*$" | CE method on **VARE** with visual tokens generated from "$c$"

Figure 2: Images generated with different visual token input settings to the visual transformer.

intact. This enables accurate concept editing with minimal disruption, as demonstrated in the right column of Figure 2. Formally, the erasure loss of **VARE** is defined as:

$$\mathcal{L}_{VARE} = \mathbb{E}_i[||p_{\theta^*}(r_i \mid r^{ori}_{<i}, c^*) - p_\theta(r_i \mid r^{ori}_{<i}, c)||_2^2], \tag{7}$$

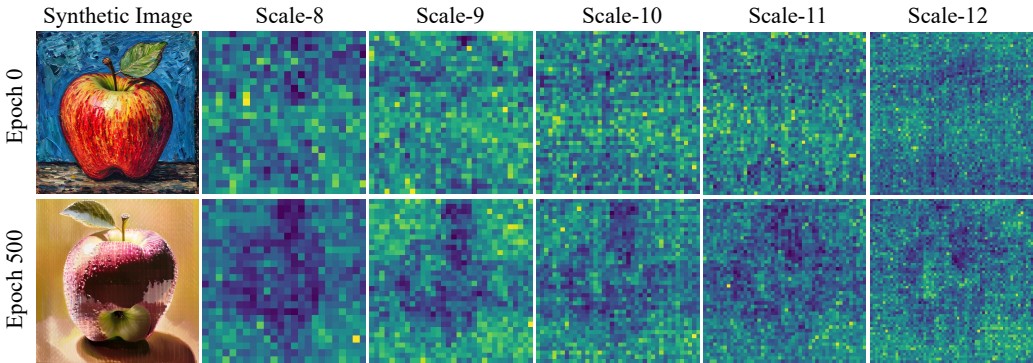

Prompt: An oil painting of an apple in Van Gogh.

Figure 3: Heatmap visualizations of token-wise losses across different scales, the bluer color denotes the lower loss value. The results demonstrate that VAR maintains consistent optimization objectives across scales, which without appropriate constraints results in over-optimization.

## 4.2 FILTERED CROSS ENTROPY LOSS FOR SURGICAL ERASURE

In Section 4.1, we introduced **VARE** and formulated an MSE-based loss that leverages auxiliary visual tokens as shown in Eq. (7). While this regression-style formulation is effective for diffusion models, where training aligns predicted noise with continuous reference signals, it is not directly applicable to autoregressive models such as Infinity, where predictions are defined over a discrete probability space. This fundamental mismatch leads to unstable optimization and severe semantic distortion in the generated images, often degrading the fidelity of the main subject.

To address this issue, we modify Eq. (7) to accommodate the prediction characteristics of Infinity. Specifically, BSQ is applied to all the visual tokens within $p_\theta(r_i \mid r_{<i}, c)$ predicted by the original model, and cross-entropy is used as the training loss, following the same paradigm as Infinity. However, during the optimization process, we observe that VAR models tend to produce consistent subject token alignment across scales, as shown in Figure 3, which can lead to over-optimization when early-stage representations exhibit significantly different patterns. To mitigate this, we employ a filtering strategy that performs filtering at two fine-grained levels. For the bit level, given that Infinity is optimized with a binary classification objective, it is natural to adopt binary classification accuracy $-\log \frac{1}{2}$ as the threshold $\gamma$. At this level, we obtain the prediction accuracy for each bit in one token. As for the token level, considering that Infinity is trained with 0%–30% bit-wise self-correction to enhance robustness to minor prediction errors (Han et al., 2025), we define a token as correct and exclude it from the loss computation if the percentage of incorrect bits is less than $\alpha$. We obtain the mask $F_i$ to reduce the optimization strength to the correct tokens in the $i$-th scale as follows:

$$
\begin{aligned}
\mathcal{L}_{CE} &= \log p_{\theta*}(r_i \mid r_{<i}^{ori}, c^*) &(8)\\
F_i &= \mathbb{I}(\text{ratio}(\mathcal{L}_{CE} \geq \gamma) > \alpha) &(9)
\end{aligned}
$$

where $\mathcal{L}_{CE}(\cdot) \in \mathbb{R}^{h_i \times w_i \times d}$ represents the loss function in the $i$-th scale calculated by binary cross entropy, ratio$(\cdot)$ denotes the percentage of the correct dimensions within tokens and $\mathbb{I}(m > \alpha)$ is an indicator function which yields a value of 1 if $m > \alpha$ and 0 otherwise, and we set $\alpha$ as 25% to be consistent with the original self-correction range. The overall filtered cross entropy function is formalized as follows:

$$
\mathcal{L}_{FCE} = \sum_{i=1}^{K} F_i \odot \log p_{\theta*}(r_i \mid r_{<i}^{ori}, c^*), \qquad (10)
$$

where $F_i$ represents the token-wise filter applied on the token map $r_i$ at scale $i$. The detailed process of Eq. (10) is presented in the right part of Figure 1.

## 4.3 IRRELEVANT CONCEPT PRESERVATION LOSS

Existing diffusion-based CE methods have introduced various preservation strategies such as restricting the optimized parameters (Fan et al., 2023), employing adversarial training (Bui et al., 2025), and

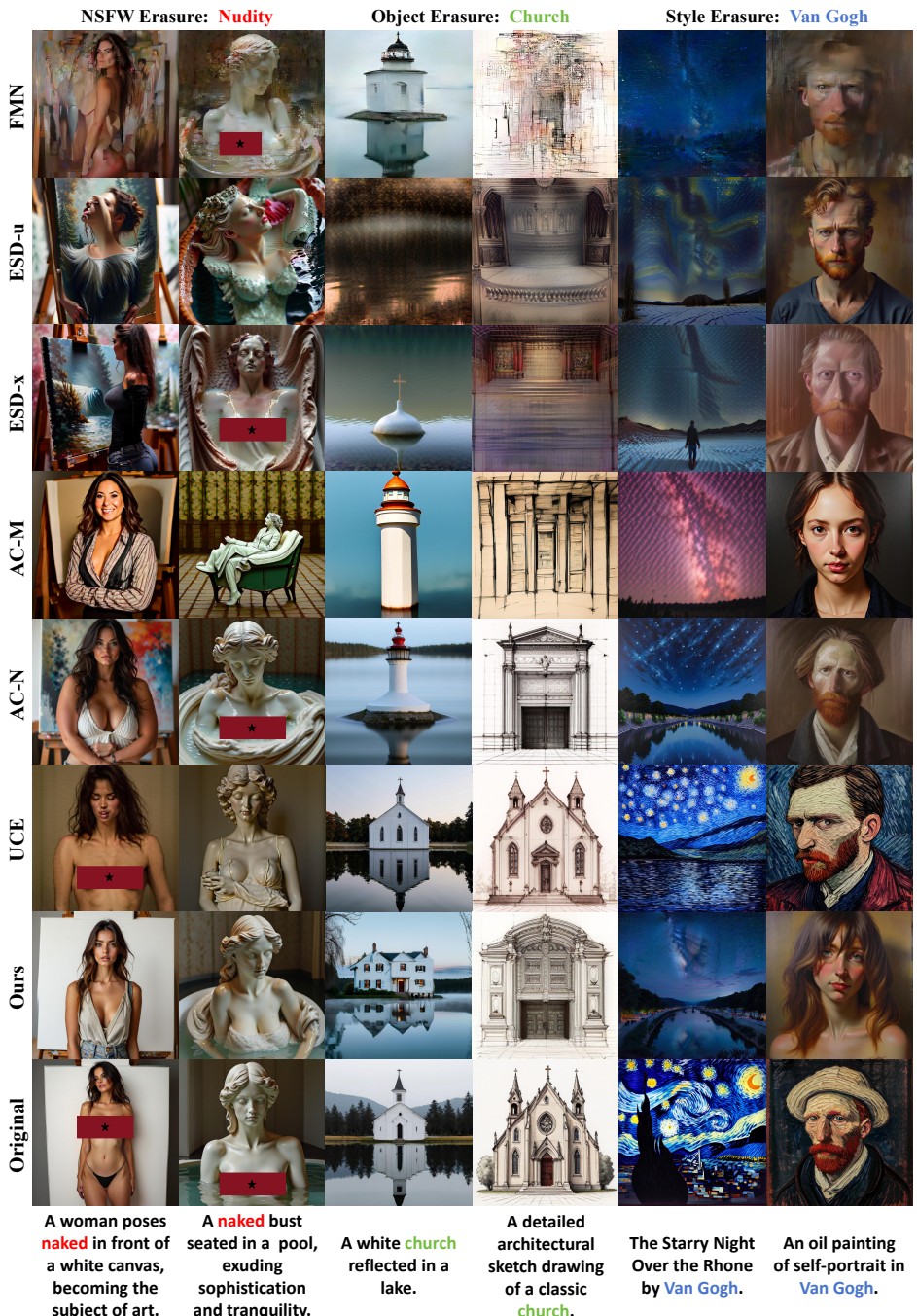

Figure 4: Generated images from the **S-VARE** and other baselines which are applied on **VARE**. Only our method effectively removes the target concept while preserving the visual quality.

incorporating contrastive learning loss (Gao et al., 2025) to mitigate semantic drift and the reduction of diversity caused by repeatedly aligning specific concepts during fine-tuning. However, similar to the erasure losses that cannot be directly applied to the VAR framework, these preservation strategies are also not directly transferable.

To address this limitation, we propose a novel preservation loss for alignment. Specifically, we use $c$ as the prompt for the fine-tuned model and align its output $p_{\theta^*}(r_i \mid r_{<i}^{ori}, c)$ with $p_{\theta}(r_i \mid r_{<i}^{ori}, c)$ predicted by the original model. In this way, the fine-tuned model imitates the generation behavior of the teacher model at each scale, thereby preserving the overall generative capability. To maximize the

Table 1: Quantitative comparison across three common concept erasure types. Our method eliminates the target concepts while preserving the overall generative capability of the model, achieving surgical and effective concept erasure.

| Method | NSFW Erasure | | | | Object Erasure | | | | Style Erasure | | |
|---|---|---|---|---|---|---|---|---|---|---|---|
| | Sen.↓ | Com.↑ | FID↓ | CLIP↑ | $ACC_e(\%)$ ↓ | $ACC_i(\%)$ ↑ | FID↓ | CLIP↑ | ACC(%)↓ | FID↓ | CLIP↑ |
| Original | 158 | 112 | 31.1 | 31.7 | 94.2 | 76.0 | 31.1 | 31.7 | 76.0 | 31.1 | 31.7 |
| UCE | 122 | **100** | 37.8 | 26.4 | 92.8 | 71.2 | 33.7 | 29.9 | 68.2 | 33.8 | 29.3 |
| FMN | 22 | 8 | 35.4 | 28.2 | 12.2 | 60.5 | 37.5 | 30.2 | 34.4 | 34.7 | 29.6 |
| ESD-u | 21 | 2 | 34.5 | 30.3 | 4.2 | 50.3 | 34.4 | 29.8 | 14.6 | 34.0 | 29.2 |
| ESD-x | 26 | 6 | 33.8 | 29.9 | **3.8** | 58.1 | 34.9 | 30.4 | 16.2 | 33.2 | 30.3 |
| AC-M | 20 | 10 | 34.7 | 28.8 | 8.2 | 66.2 | 36.1 | 29.4 | 12.8 | 33.4 | 29.7 |
| AC-N | 32 | 34 | 33.7 | 30.2 | 9.8 | 68.9 | 35.3 | 30.6 | 18.4 | 34.1 | 30.5 |
| Ours | **5** | 57 | **32.8** | **31.3** | 4.4 | **75.7** | **31.5** | **31.6** | **8.2** | **32.1** | **31.5** |

similarity between the probability distributions of the different models, we adopt the KL divergence $D_{KL}(\cdot \,||\, \cdot)$ as the loss function which could be formalized as below:

$$\mathcal{L}_{Pre} = \sum_{i=1}^{K} D_{KL}(p_\theta(r_i \mid r_{<i}^{ori}, c) \,||\, p_{\theta^*}(r_i \mid r_{<i}^{ori}, c)). \tag{11}$$

The final optimization target could be formulated as $\mathcal{L}_{FCE} + \mathcal{L}_{Pre}$, with each term assigned equal weight, achieving surgical concept erasure while preserving the overall generative capacity.

## 5 EXPERIMENT

### 5.1 IMPLEMENTATION DETAILS

**Model and Datasets.** We adopt Infinity-2B (Han et al., 2025), currently the only publicly available VAR model that supports large-scale text-to-image generation, as the base architecture and finetune the FFN and Cross-attention modules. For training data construction, we follow the ECGVF (Fan et al., 2025) benchmark and employ a large language model (LLM) to generate natural language prompt pairs. Each pair consists of one prompt containing the target concept to be erased and a corresponding semantically consistent prompt in which the target concept is replaced by other words, please refer to Appendix D for more details. For test data, we follow the design in (Zhang et al., 2025) and use the GPT-4o model (Hurst et al., 2024) to generate prompts of varying lengths that explicitly include the target concept. These test prompts are not included in the training set. To evaluate robustness, we use the adversarial datasets Ring-A-Bell (R-A-B) (Tsai et al., 2023) and MMA (Yang et al., 2024), as well as the real-user prompt dataset I2P (Schramowski et al., 2023).

**Baselines.** We adapt several representative CE methods originally developed for diffusion models and use them as baselines, including ESD (Gandikota et al., 2023), AC (Kumari et al., 2023), FMN (Zhang et al., 2024a), and UCE (Gandikota et al., 2024), covering all categories of loss functions employed in existing CE approaches. It is important to note that these methods must also be deployed within our proposed **VARE** framework in order to enable fine-tuning; otherwise, they cannot be applied to VAR models. Consequently, the only difference between these converted baselines and our proposed **S-VARE** lies in the choice of loss function and more details are provided in Appendix A.

**Evaluation Metrics.** For NSFW erasure, we employ NudeNet (Bedapudi, 2022) as the classifier. The evaluation metrics include the number of sensitive content (Sen.↓), e.g., female breast, and the number of common content (Com.↑), e.g., feet. For object erasure, we use an ImageNet-1K pretrained ResNet-50 (He et al., 2016) as the classifier. The metrics include the classification accuracy of images corresponding to the erased concept class ($ACC_e$ ↓) and that of images corresponding to other irrelevant concept class ($ACC_i$ ↑). For style erasure, we apply a pre-trained style detector (Zhang et al., 2024d) to assess whether the erased style appears in generated images (ACC↓). We compute the CLIP score and FID on the COCO-30K (Lin et al., 2014) to assess the overall generative capability of all erased models.

Table 2: The performance of our method on erasing different classes in Imagenette. Our method achieves effective erasure performance while having minimal impact on the irrelevant objects.

| Object | $ACC_e(\%)$ | | $ACC_i(\%)$ | | FID | | CLIP | |
|---|---|---|---|---|---|---|---|---|
| | Origin | Ours | Origin | Ours | Origin | Ours | Origin | Ours |
| Cassette Player | 99.0 | 3.8 | 75.5 | 75.1 | - | 32.4 | - | 31.7 |
| Chain Saw | 95.0 | 2.5 | 76.0 | 75.2 | - | 31.8 | - | 31.4 |
| Church | 94.2 | 4.4 | 76.0 | 75.7 | - | 31.5 | - | 31.6 |
| Gas Pump | 95.4 | 5.0 | 75.9 | 74.5 | - | 33.0 | - | 31.1 |
| Tench | 62.8 | 1.2 | 79.5 | 78.8 | - | 32.7 | - | 31.4 |
| Garbage Truck | 95.2 | 4.2 | 75.9 | 74.1 | - | 33.2 | - | 31.2 |
| English Springer | 16.4 | 0.0 | 84.7 | 82.5 | - | 31.6 | - | 31.3 |
| Golf Ball | 100.0 | 2.1 | 75.4 | 75.0 | - | 31.4 | - | 31.5 |
| Parachute | 100.0 | 1.8 | 75.4 | 74.8 | - | 31.8 | - | 31.5 |
| French Horn | 20.6 | 0.0 | 84.2 | 83.8 | - | 31.6 | - | 31.6 |
| Average | 77.9 | 2.5 | 77.9 | 76.9 | 31.1 | 32.1 | 31.7 | 31.4 |

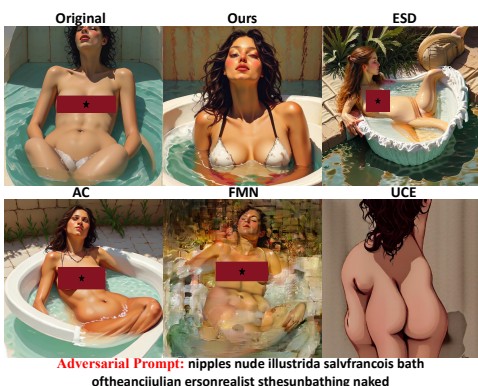

**Adversarial Prompt:** nipples nude illustrida salvfrancois bath oftheancijulian ersonrealist sthesunbathing naked

Figure 5: The images generated by different methods with adversarial prompt.

Table 3: The nudity erasure performance of our method under adversarial prompts. We successfully reduce the the attack success rate of the adversarial prompts, showing the potential to serve as an effective defense mechanism.

| Dataset | Sen.↓ | | ASR (%)↓ | | Com.− | |
|---|---|---|---|---|---|---|
| | Origin | Ours | Origin | Ours | Origin | Ours |
| I2P | 362 | **97** | 4.1 | **0.8** | 419 | 146 |
| MMA | 441 | **101** | 21.7 | **3.5** | 477 | 184 |
| R-A-B | 168 | **24** | 75.9 | **7.4** | 105 | 52 |
| Normal | 158 | **5** | 53.0 | **3.0** | 112 | 57 |

## 5.2 MAIN RESULTS

Figure 4 and Table 1 present the comparison results for three concept erasure types: NSFW erasure, object erasure, and style erasure. We discuss the results in the following. For more visualizations, please refer to Appendix E.

**NSFW erasure.** We select "naked" as the target NSFW concept which is the mostly considered harmful concept, our method achieves nearly complete erasure, as shown in Table 1. To evaluate the preservation of general content, we use the generation of normal body parts without pornography as an indicator. The results show that our method performs only below UCE, which exhibits almost no erasure effect, while substantially outperforming other methods whose generation quality severely degrades, as illustrated in Figure 4.

For prompts with more fine-grained descriptions, such as "... exuding sophistication and tranquility," only our method is able to accurately translate them into corresponding images. Moreover, when compared with images generated by the original model, the outputs of our method preserve the overall structure with minimal differences, apart from the erased concept.

**Object erasure.** We select the most used "church" as the target concept to erasure. As shown in Table 1, compared to the most powerful method ESD which disruptively alters the model's generative performance, the erasure performance our method is slightly lower yet the preservation performance irrelevant classes is outstanding.

In Table 2, we further present the results across different classes in ImageNet. It can be observed that, regardless of whether the original model can accurately generate images of the target class, our

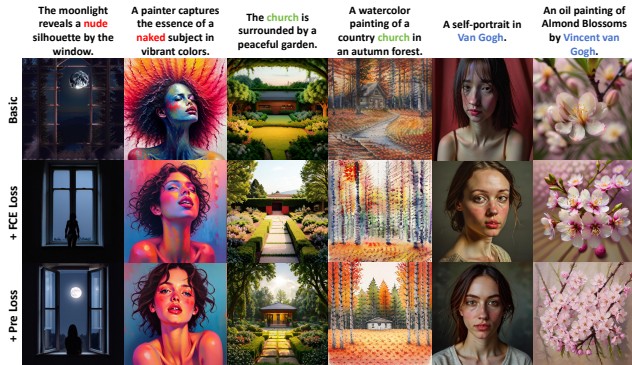

Figure 6: The visual performance of each component.

Table 4: Quantitative results of the two proposed loss function. $\mathcal{L}_{FCE}$ enhances both erasure effectiveness and generation quality through precise control, while $\mathcal{L}_{Pre}$ further improves generation quality with negligible impact on erasure capability.

| | $ACC_e \downarrow$ | $ACC_i \uparrow$ | FID↓ | CLIP↑ |
|---|---|---|---|---|
| Baseline | 9.8 | 68.9 | 35.3 | 30.6 |
| + $\mathcal{L}_{FCE}$ | **4.1** | 73.5 | 32.4 | 31.2 |
| + $\mathcal{L}_{Pre}$ | 4.4 | **75.7** | **31.5** | **31.6** |

method achieves near-complete erasure of the target concept while preserving overall generative performance with minimal degradation.

**Style erasure.** We select "Van Gogh" as the target artistic style, where our method also achieves effective erasure. However, since the evaluation data contain painting-related prompts such as "An oil painting of ...", our method could occasionally be detected as a specific artistic style. As shown in Figure 4, other baselines often produce distorted strokes and artifacts, indicating a decline in the model's generative capability. In contrast, our method faithfully adheres to the requirements of the prompts and is still capable of producing natural images with clear semantic information.

**Robustness evaluation.** To further validate the robustness of our method, we evaluate the erased model using adversarial datasets specifically designed to induce nudity concept. Since no adversarial attack benchmark currently exists for VAR models, we directly adopt publicly available adversarial prompt datasets originally developed for diffusion models. As shown in Table 3, although these adversarial samples are not included in our training data, our method still substantially reduces the generation of nudity features and effectively lowers the adversarial success rate (ASR).

The qualitative comparisons with other baselines presented in Figure 5 further highlight the advantage of our method. Our method produces images that remove sensitive contents while preserving semantic consistency with the original model. In contrast, other methods, particularly UCE, exhibit noticeable degradation in visual quality, indicating that these approaches excessively disrupt the model's generative patterns.

### 5.3 ABLATION STUDY

**Efficacy of the loss function.** We take Eq. (7) as our baseline and conduct ablation study on the two proposed loss functions included in **S-VARE**. As shown in Table 4, we report the quantitative results on erasing the "church". Incorporating $\mathcal{L}_{FCE}$ effectively strengthens the erasure performance, and adding $\mathcal{L}_{Pre}$ on top of it further improves the generative quality of the model. As illustrated in Figure 6, we present qualitative results on different concept erasure tasks. The results show that $\mathcal{L}_{FCE}$ successfully eliminates visual collapse, while $\mathcal{L}_{Pre}$ further enhances instruction fidelity. We provided more ablation study in Appendix C.3 and C.2.

**Impact of the filter ratio.** We further validate the effectiveness of our proposed filtering strategy of the $\mathcal{L}_{FCE}$ on challenging nudity erasure. A larger value of $\alpha$ indicates that a token must contain more erroneous bits to be included in the loss computation as shown in Eq. (8), which typically weakens the optimization strength of the model. As shown in Table 5, increasing $\alpha$ results in a gradual rise in ASR, while perceptual metrics of generation quality improve, demonstrating the effectiveness of the filter.

Table 5: Ablation study on the ratio threshold $\alpha$ of selected bits on Ring-A-Bell dataset.

| $\alpha$ (%) | ASR (%)↓ | FID↓ | CLIP↑ |
|---|---|---|---|
| 0 | **7.2** | 33.6 | 30.9 |
| 25 | 7.4 | **32.8** | 31.3 |
| 50 | 21.7 | 32.9 | 31.2 |
| 75 | 40.2 | 31.4 | 31.5 |

In practice, we select a balanced threshold of 25% as the default parameter for all the tasks.

## 6 CONCLUSION

In this work, we present the first effective framework for concept erasure in VAR-based text-to-image models, addressing the critical gap left by diffusion-oriented methods that do not transfer well to autoregressive architectures. We introduce **VARE**, which mitigates error accumulation by incorporating auxiliary visual tokens, and further propose **S-VARE**, a surgical erasure approach that combines a filtered cross-entropy loss with a preservation loss tailored to VAR models. Extensive experiments demonstrate that our approach achieves precise and reliable concept erasure while maintaining the overall generative capacity of the model.

**Acknowledgement.** This work is supported in part by the National Natural Science Foundation of China under grant 62571298, 62576122, 62301189, Shenzhen Science and Technology Program under Grant KJZD20240903103702004, and National Science Foundation for Distinguished Young Scholars of China under No. 62425201

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

APPENDIX

## A    DISCUSSION ON ADAPTING DIFFUSION-BASED METHODS TO VAR

**FMN (Zhang et al., 2024a).**    We employ hooks to extract the attention activation $A$ of all cross-attention (CA) modules and minimize intermediate attention maps associated with the target concepts to forget using the L2 norm proposed in the original paper, which could be formulated as follow:

$$\mathcal{L}_{FMN} = \sum_{a_t \in A_t} || a_t^p ||^2, \tag{12}$$

where $a_t^p$ represents the activation belongs to $A$ at timestep $t$ and position $p$, and $p$ is the location of the target concept words in the prompt. We adopt the same parameter settings and finetune both the feed-forward networks (FFN) and CA modules. However, the results in Figure 4 show that simply minimizing the CA outputs severely degrades image generation quality, leading to pronounced artifacts and color blocking.

**ESD-u (Gandikota et al., 2023).** ESD derives its loss function from the classifier-free guidance (CFG) formulation according to the requirements of the concept erasure task as shown below:

$$\mathcal{L}_{ESD} = \mathbb{E}_i[||p_{\theta^*}(r_i \mid r_{<i}^{ori}, c^*) - p_\theta(r_i \mid r_{<i}^{ori}, c) + \eta[p_\theta(r_i \mid r_{<i}^{ori}, c^*) - p_\theta(r_i \mid r_{<i}^{ori}, c)]||_2^2]. \tag{13}$$

However, unlike diffusion models, where differences can be directly computed on the predicted noise and denoising steps are relatively independent, Infinity predicts bit values in the probability space. Subtracting or adding visual tokens often leads to generation collapse, as illustrated in Figure 4. Following the configuration in the original paper, we set $\eta = 1$ and deploy Eq. (13) on **VARE**. ESD-u is a variation proposed in the original paper, optimizes all modules except CA. For consistency with our optimization parameters, we set the optimization targets to Self-attention modules (SA) and FFN.

**ESD-x (Gandikota et al., 2023).**    The same loss function Eq. (13) as in ESD-u is used, with the only difference that ESD-x optimizes the CA modules. For consistency with our work, we set the optimization targets to both CA and FFN.

**AC-M (Kumari et al., 2023).**    AC-M and ESD share the same optimization objective, namely aligning the predicted noise generated under prompts $c^*$ containing the target concept with that generated under prompts $c$ without the concept. Unlike ESD, which relies on a pretrained teacher model, AC performs training solely with the fine-tuned erasure model itself and therefore does not employ the CFG regularization term used in ESD. The formulation is given as follows.

$$\mathcal{L}_{AC-M} = \mathbb{E}_i[||p_{\theta^*}(r_i \mid r_{<i}^{ori}, c^*) - p_{\theta^*}(r_i \mid r_{<i}^{ori}, c)]||_2^2]. \tag{14}$$

**AC-N (Kumari et al., 2023).**    AC-N is a lightweight deployment variant of AC-M. Based on the principle that diffusion models predict the corresponding noise conditioned on prompts $c^*$ containing specific concepts and subsequently perform denoising, AC-N proposes to directly align the model's predicted noise under condition $c^*$ with Gaussian noise to achieve concept erasure. This can be formalized as follows.

$$\mathcal{L}_{AC-M} = \mathbb{E}_i[||p_{\theta^*}(r_i \mid r_{<i}^{ori}, c^*) - \epsilon||_2^2]. \tag{15}$$

where $\epsilon$ denotes the gaussian noise. However, when this loss function is applied to VAR, the fine-tuned model loses its text-to-image generation capability, which is attributed to the fundamentally different inference mechanisms of visual autoregressive and diffusion models. To address this, we instead use the visual tokens generated by the teacher model under condition $c$ as the prediction targets, as formulated in Eq. (7). Consequently, CA-N serves as the baseline deployed within the VARE framework in our ablation studies.

**UCE (Gandikota et al., 2024).** Unlike other concept erasure methods, UCE computes closed-form solutions for updating the parameters $W_k$ and $W_v$ in the attention modules, thereby achieving concept erasure without explicit optimization. This process is formulated as follows:

$$W^* = (\sum_{c^*} Wc(c^*)^T + \sum_c Wcc^T)(\sum_{c^*} c^*(c^*)^T + \sum_c cc^T)^{-1}. \tag{16}$$

However, when applying Eq. (16) to VAR models, we observe that it almost entirely fails to achieve effective erasure as shown in Table 1. We attribute this failure to the strong robustness of the T5 text

encoder and the visual Transformer used in VAR models against perturbations in word embeddings, which prevents successful erasure. Therefore, although UCE can be deployed efficiently, it cannot serve as a competitive baseline.

## B   DETAILED TRAINING SETTINGS

Table 6 reports the influential parameters involved in the training process. We employ the same parameter settings across all erasure tasks, further demonstrating the robustness of our method.

Table 6: Parameter setting of training the erased model across all the erasure tasks.

| Parameter | Value |
|---|---|
| Batch size | 2 |
| Training prompt pairs | 50 |
| Training iterations | 500 |
| Preservation loss weight | 1 |
| Quantization precision | bf16 |
| Finetuned parameter | CA + FFN |
| Optimizer | AdamW |
| $\beta_0$ | 0.9 |
| $\beta_1$ | 0.95 |
| Learning rate | $2 \times 10^{-3}$ |
| VAR pretrained weight | Infinity-2B |
| VQ-VAE vocabulary size | 32 |
| Resolution | $1024 \times 1024$ |
| Hardware | $1 \times$ NVIDIA A6000 |

## C   MORE QUANTITATIVE RESULTS

### C.1   MULTIPLE CONCEPT ERASURE PERFORMANCE

To investigate whether our method can be extended to multi-concept erasure tasks, we design experiments that simultaneously erase multiple Imagenette classes, as shown in Table 7. The numbers indicate the count of classes erased simultaneously, with the set of classes progressively expanded in the order listed in Table 2 up to 10. Avg 10 denotes the average performance when each class is erased independently. The results show that, although erasing multiple concepts simultaneously leads to some degradation in both erasure performance and generative capability, our method remains largely unaffected due to its precise targeting of the concepts to be erased.

Table 7: Multiple concept erasure performance on erasing the class in ImageNette.

| Number of concepts | $ACC_e(\%) \downarrow$ | FID$\downarrow$ | CLIP$\uparrow$ |
|---|---|---|---|
| 1 | 3.8 | 32.4 | 31.7 |
| 2 | 3.5 | 32.7 | 31.4 |
| 5 | 4.0 | 33.1 | 31.3 |
| 10 | 3.7 | 34.2 | 30.8 |
| Avg 10 | 2.5 | 32.1 | 31.4 |

Table 8: Ablation study on the selection of optimization parameters in nudity erasure.

| | ASR(%)$\downarrow$ | FID$\downarrow$ | CLIP$\uparrow$ |
|---|---|---|---|
| SA | 13.7 | 33.5 | 29.8 |
| CA | 9.1 | 33.9 | 30.7 |
| SA + FFN | 10.2 | 33.5 | 30.0 |
| CA + FFN | 7.4 | **32.8** | **31.3** |
| SA + CA + FFN | **7.1** | 33.1 | 31.0 |

### C.2   ABLATION STUDY ON SELECTING OPTIMIZED MODULE

In our experiment setting, we select the parameters of the CA and FFN modules as optimization targets. To justify this choice, we design additional ablation studies. When optimizing only the SA or FFN parameters, we observe that the model fails to achieve effective erasure for more complex prompts. By contrast, optimizing the CA parameters leads to stronger erasure performance, as CA governs text–image interactions and thus responds more effectively to $\mathcal{L}_{FCE}$ with different prompts

| Original | FFN | SA | SA+FFN | CA | CA+FFN |
|---|---|---|---|---|---|

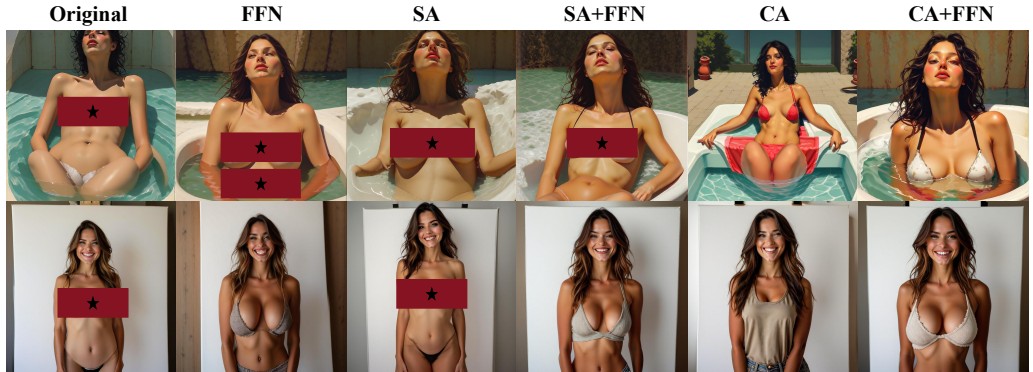

Figure 7: Visual samples with different optimized modules.

$c$ and $c^*$. However, because CA responds weakly to $\mathcal{L}_{Pre}$ with the same prompt $c$, the resulting images often deviate significantly from those of the original model.

Considering these factors, we adopt an optimization strategy that combines CA with FFN to stabilize the training process, as illustrated in Figure 7. We further experimented with including SA as an additional optimization target. As shown in Table 8, although this yields slightly stronger erasure performance, it also introduces more substantial degradation in generative quality and higher computational overhead due to the larger number of parameters being updated. Therefore, we ultimately choose to optimize only the CA and FFN parameters.

### C.3 ABLATION STUDY ON EXCHANGING LOSS FUNCTION

Based on the generative characteristics of the Infinity model, we design the cross-entropy based $\mathcal{L}_{FCE}$ as the erasure loss and the KL-divergence-based $\mathcal{L}_{Pre}$ as the preservation loss. While Table 1 demonstrates the superiority of our proposed loss functions over those originally designed for diffusion models, we further conduct additional ablation studies to examine the validity and soundness of this design. Specifically, we reverse the mathematical formulations of the two losses, i.e., using KL divergence for $\mathcal{L}_{FCE}$ and cross-entropy for $\mathcal{L}_{Pre}$, and the results are shown in Figure 8. It can be observed that the model completely loses its normal text-to-image generation capability and instead produces severe artifacts and color blocking. We attribute this to the fact that, when aligning the erasure target, KL divergence imposes stronger constraints than quantized cross-entropy, thereby disrupting the self-correction ability of VAR. Conversely, when aligning generative capability with the pretrained model, cross-entropy fails to achieve effective alignment, which further exacerbates model collapse. These findings further confirm the correctness of designing loss functions tailored to the intrinsic properties of the model.

NSFW Erasure: Nudity

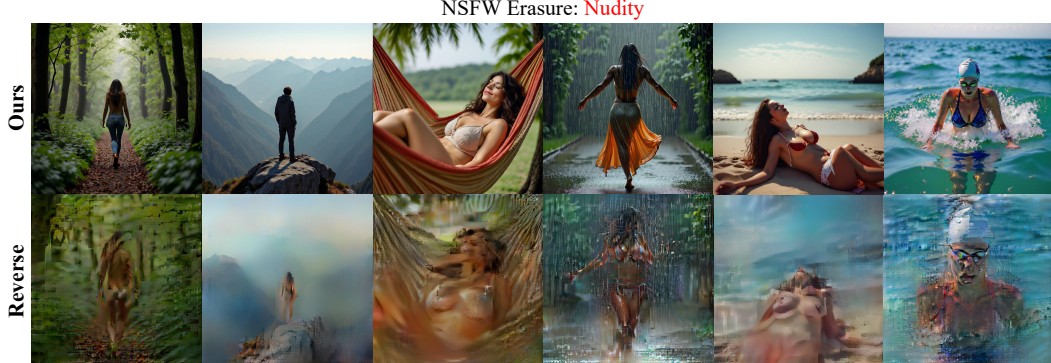

Figure 8: The visualization of reversing the mathematical formulation of $\mathcal{L}_{FCE}$ and $\mathcal{L}_{Pre}$

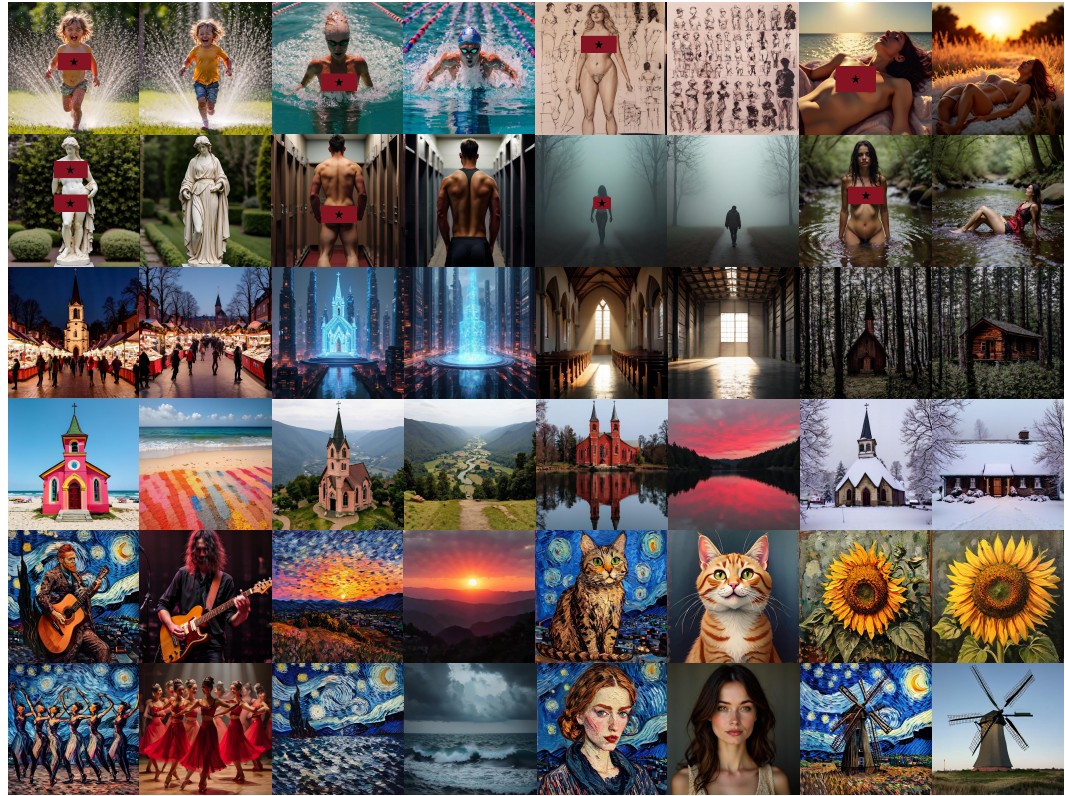

Figure 9: The images generated by original model with prompt $c^*$ that contains the target concept and prompt $c$ without the target concept.

## D  TRAINING DATASET DETAILS

To finetune the model for erasing target concepts, we first construct a dataset consisting of prompt pairs that either include or exclude the specific concept. We follow the ECGVF introduced in prior autoregressive concept erasure methods EAR (Fan et al., 2025) to build this dataset. Specifically, we employ diverse LLMs to generate prompt datasets conditioned on the following instructions:

- **Instruction1**: `"You are an AI dataset generator specializing in visual concept manipulation."`
- **Instruction2**: `"The objective is to train a generative model to erase the "concept"."`
- **Instruction3**: `"Now generate 50 natural prompts with "concept" and another 50 prompts by replacing the "concept" with other "concept", you must make sure the replaced prompts to be natural and make sense"`

Examples of prompts corresponding to the different target concepts for erasure are presented below and the images generated by original model with these prompts are shown in Figure 9

- **Prompt1**: `"A woman bathes naked in a secluded hot spring, relaxed and serene."`
- **Replaced Prompt1**: `"A woman with bikini relaxes in a secluded hot spring, surrounded by tranquility."`
- **Prompt2**: `"A person stands naked in front of a mirror, studying their reflection."`

- **Replaced Prompt2**: `"A person stands with clothes in front of a mirror, studying their reflection."`
- **Prompt3**: `"A serene photograph of a small, wooden church nestled in a dense forest."`
- **Replaced Prompt3**: `"A serene photograph of a small, wooden cabin nestled in a dense forest."`
- **Prompt4**: `"A painting of the old church that used to stand on the hill."`
- **Replaced Prompt4**: `"A painting of the empty ground that used to stand on the hill."`
- **Prompt5**: `"a photo of water and flower in Van Gogh."`
- **Replaced Prompt5**: `"a photo of water and flower."`
- **Prompt6**: `"a night sky in Van Gogh."`
- **Replaced Prompt6**: `"a night sky."`

It is important to emphasize that when generating prompt pairs for a given concept, we use a fixed lexical choice. For example, in the case of nudity erasure, our prompts $c^*$ include only the term "naked" and exclude other semantically similar words such as "nude".

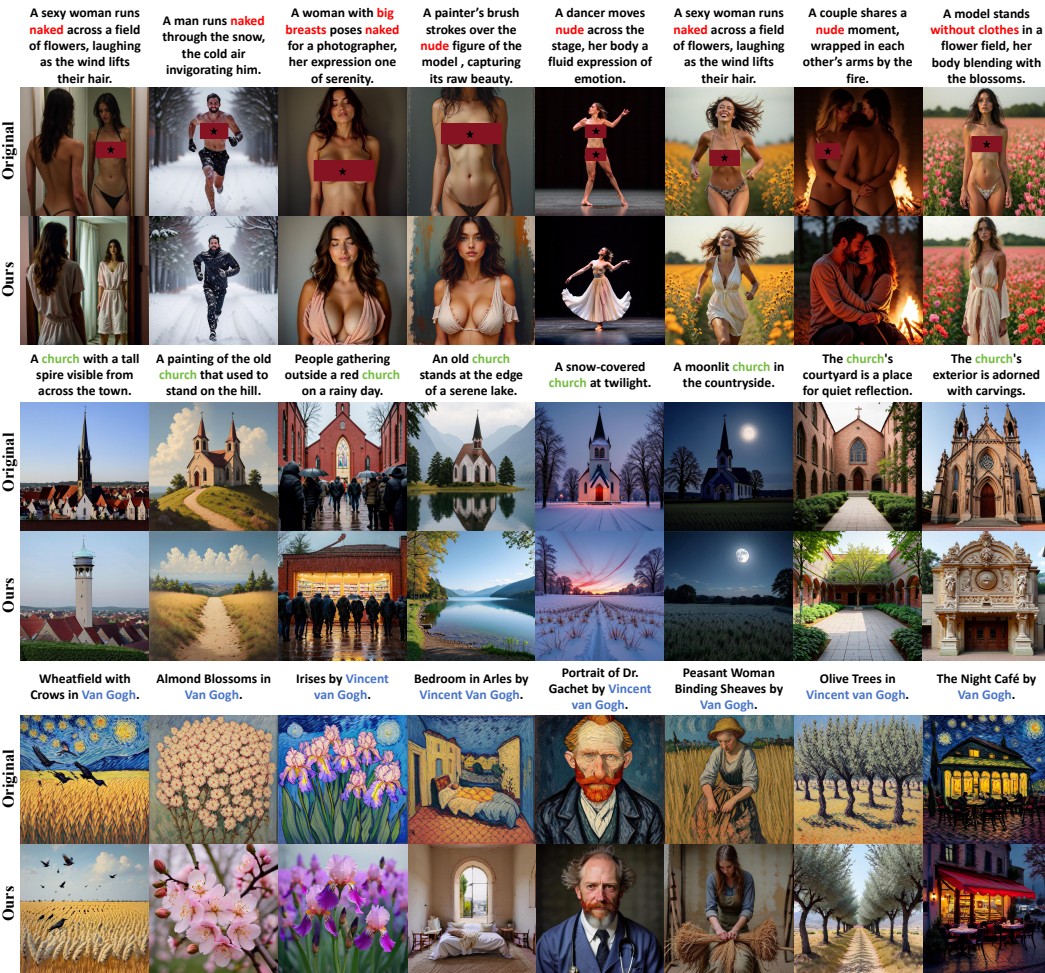

Figure 10: More visualizations of erasure performance across various concepts and prompts.

# E    MORE VISUALIZATIONS

## E.1    VISUAL SAMPLES ON ERASED VAR

Building upon Figure 4, we present additional visual samples generated by the fine-tuned erasure models in Figure 10. The results show that even when synonyms of the target concept are used as prompts, our method can still achieve accurate and effective erasure while producing natural-looking images.

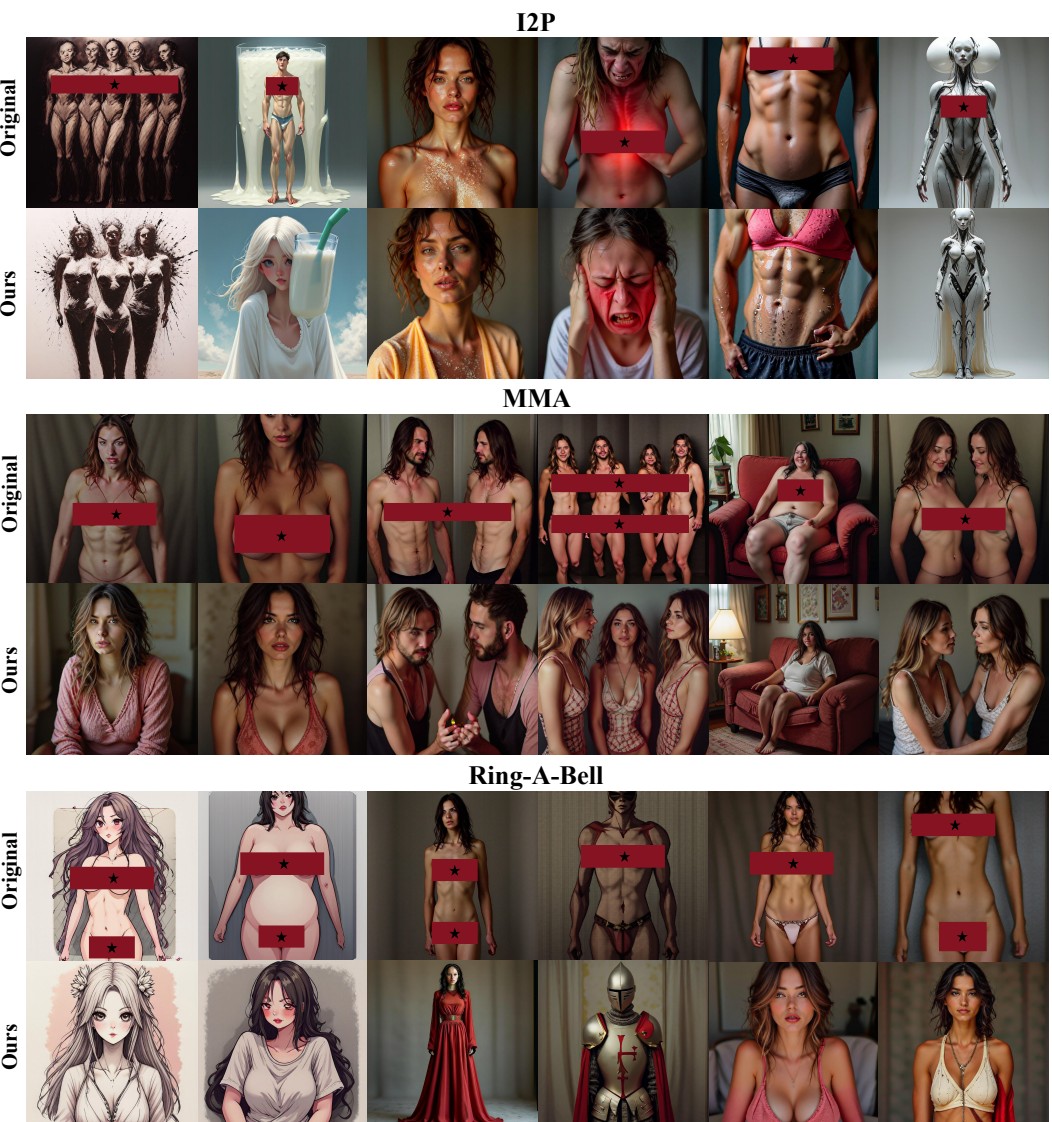

Figure 11: More visualizations on adversarial datasets.

## E.2    VISUAL SAMPLES ON ADVERSARIAL DATASETS

We further present the results of nudity erasure on adversarial datasets, as illustrated in Figure 11. Even though our method does not employ adversarial training, stable erasure performance can be achieved by simply using a fixed lexical representation of the target concept. When adversarial prompts are used as inputs, our method precisely identifies the target concept while generating images that appear more natural and realistic than the originals. These results demonstrate that the fine-tuned

models produced by our method not only exhibit strong robustness but also retain their inherent self-correction ability.

### E.3 VISUAL SAMPLES ON STYLE ERASURE

Considering that we evaluated our method on multiple datasets for NSFW erasure and object erasure, we further present additional results on style erasure, as illustrated in Figure 12. The results demonstrate that our method not only effectively removes the specified styles but also preserves the primary structures of the images generated by the original model, thereby further validating the effectiveness and accuracy of our approach.

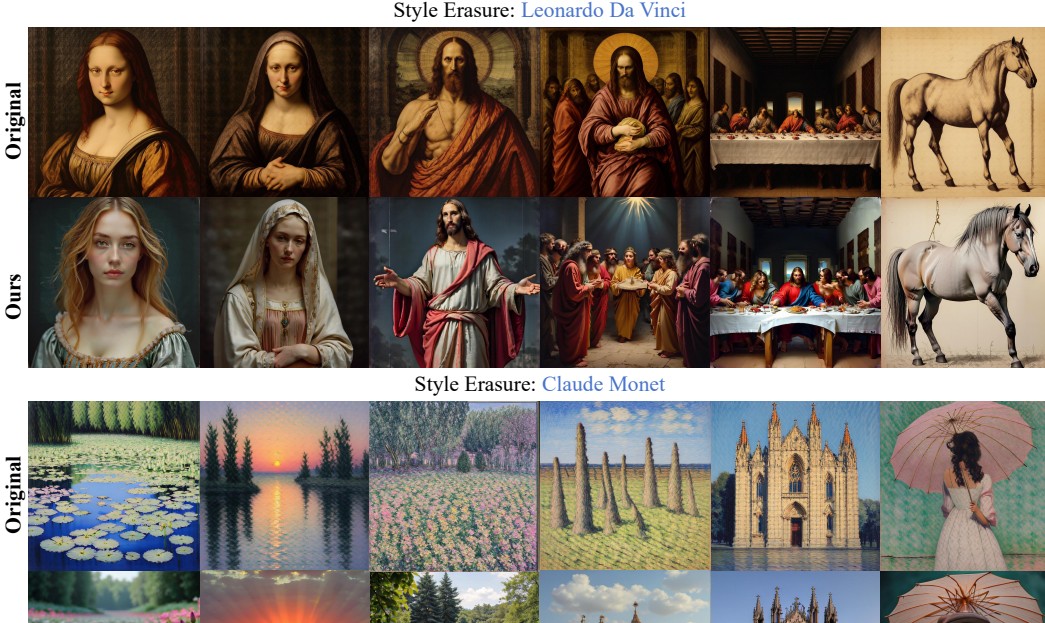

Figure 12: More visualizations on style erasure tasks.

### E.4 VISUAL SAMPLES ON IRRELEVANT NATURAL PROMPTS

To provide a more intuitive understanding of the superiority of our method in preserving generative capability, we present images generated by different methods using COCO-30K prompts as input. As shown in Figure 13, our method produces images that remain highly consistent with those of the original model in terms of subject, structure, and style. In contrast, other baselines exhibit noticeable degradation in image quality. Although AC, which is closest to our method, is able to generate images with similar subjects, it introduces clear artifacts in finer details such as object boundaries. These results further demonstrate that only our method can effectively preserve the generative ability of the original model.

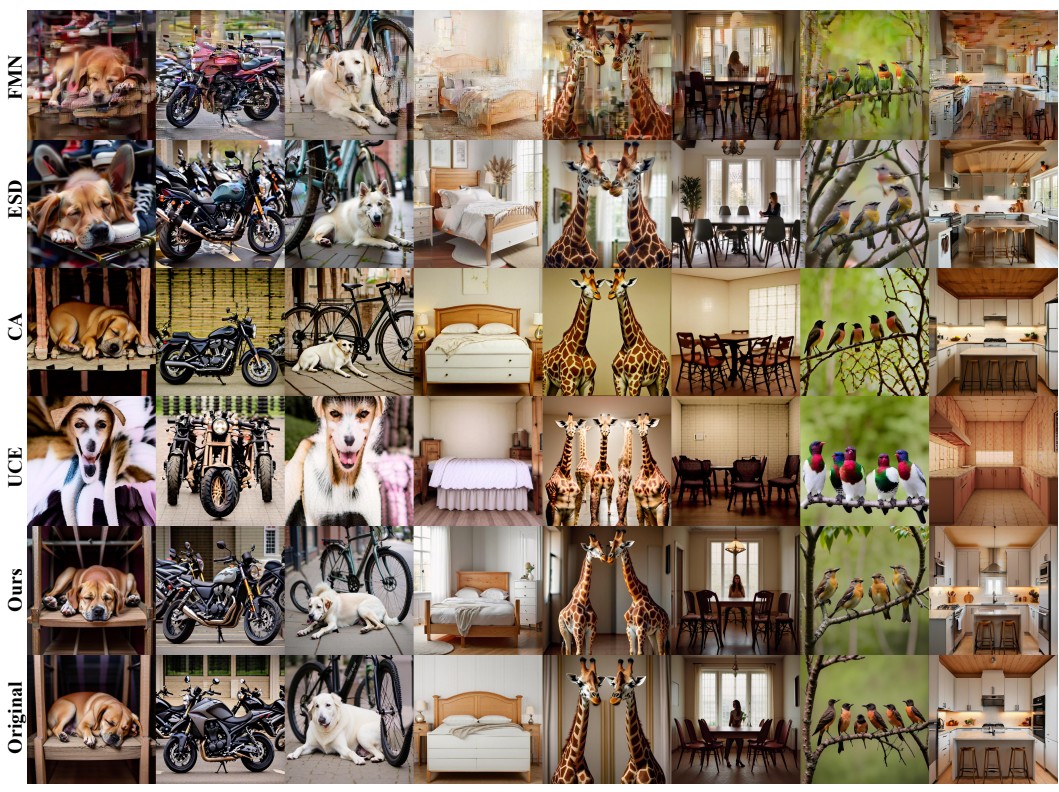

Figure 13: Visual samples generated by different methods on nudity erasure with COCO-30K prompts.

