# OpenReview forum: "Closing the Safety Gap: Surgical Concept Erasure in Visual Autoregressive Models"
_ICLR.cc/2026/Conference — ICLR 2026 Poster_

### Official Review · Reviewer_TYDT · 2025-10-30

**Soundness:** 3
**Presentation:** 3
**Contribution:** 3
**Rating:** 6
**Confidence:** 4

**Summary:**

This paper proposes a new concept erasure framework for visual autoregressive models (VAR), called VARE, and a fine-grained method S-VARE. The authors point out that existing concept erasure methods designed for diffusion models do not work well in autoregressive models due to error accumulation across scales. To solve this, they use auxiliary visual tokens during training to stabilize optimization and introduce two new loss functions: a filtered cross-entropy loss to remove only harmful tokens and a preservation loss to keep unrelated content unchanged. Experiments show strong results on NSFW, object, and style erasure tasks.

**Strengths:**

1.The paper first proposes a concept erasure method specifically for VAR models.

2.The use of auxiliary visual tokens during training is a practical and effective way to reduce error accumulation in autoregressive generation.

**Weaknesses:**

1. The method is designed for single-concept erasure and does not support removing multiple concepts at once, which limits its applicability in real-world scenarios where multiple harmful or unwanted concepts may co-occur.

2. The paper lacks a token-level interpretability analysis. It is unclear which tokens are being erased during training, at which scales, or how the filtering mechanism behaves across different types of concepts.

3. The token filtering mechanism uses a fixed threshold for all tokens without considering their semantic relevance to the target concept. A more adaptive approach could improve precision by focusing only on tokens that are highly correlated with the concept to be erased.

4. Missing related works:

[1] Concept corrector: Erase concepts on the fly for text-to-image diffusion models

[2] Dark miner: Defend against unsafe generation for text-to-image diffusion models

[3] One Image is Worth a Thousand Words: A Usability Preservable Text-Image Collaborative Erasing Framework

[4] Erasing More Than Intended? How Concept Erasure Degrades the Generation of Non-Target Concepts

**Questions:**

1. Can the method be extended to erase multiple concepts at the same time?

2. Can this method be applied to other autoregressive models beyond Infinity?

3. I’m unsure how the filtered cross-entropy loss functions in the context of erasing artistic styles. For object erasure, it makes intuitive sense to optimize the "object-related" visual tokens, as they directly correspond to the target object in the image. However, when it comes to styles, the token-wise loss seems to be spread across the entire image, as shown in the first row of Figure 3, where the concept of "blue" appears throughout the image. In such cases, how does the filtered cross-entropy loss effectively work?

---

> ### Author Response · Authors · 2025-11-19
>
> We thanks for your precious effort in providing suggestions to enhance the quality of our paper. We have provided detailed point-by-point responses as follows. We hope these responses can adequately address your concerns.
>
> >**Weakness 1 & Question 1: The experimental results of multiple concept erasure.**
>
> We provide experiments on multi-concept erasure in **Appendix C.1**. The results in **Table 7** show that although our method exhibits some performance degradation when erasing multiple concepts simultaneously, it still effectively removes all target concepts while preserving the model’s generative capability.
>
> >**Weakness 2: The token-level interpretabe analysis of the proposed method.**
>
> Thank you for your suggestion. In **Figure 3**, we provide an example of the variation in loss function values of different tokens when erasing the style concept. After optimization, tokens at different scales effectively reduce the discrepancy with the target token, showing consistent optimization trends across all scales. We will include additional visual examples of token value differences before and after optimization in the final version.
>
> >**Weakness 3: The design of the token filtering mechanism.**
>
> In deploying the filtering mechanism, we first calculate the proportion of erroneous bits in each token based on the loss function value and optimize only the tokens whose error rate exceeds a predefined threshold. In this process, **the loss function value serves as a quantitative indicator closely tied to the concept to be erased, guiding the tokens that need further alignment with the target concept at the semantic level**.
>
> Additionally, we present ablation experiments on the threshold $\alpha$ in **Table 5**. The results show that our method achieves a good balance between erasure efficiency and generative capability without requiring additional optimization. In future work, we will explore how to use more advanced metrics to measure the semantic differences between tokens and implement more precise constraints. We also welcome further discussion with you on this issue.
>
> >**Weakness 4: More related works.**
>
> Thanks for your feedback. We will include these recent works and citations in the related work section.
>
>
> >**Question 2: Experimental results on other visual autoregressive model.**
>
> Our work addresses the issue of sensitive concepts generated by text-to-image **visual autoregressive models**, such as Infinity. Since Infinity is currently the only model capable of large-scale text-to-image tasks, we validated our method on Infinity-2B. Following your suggestion, we also conducted experiments on the original **VAR-d30** model, which uses VQ-VAE and only supports label-based generation on the ImageNet-1K dataset. For object concept erasure, we implemented the **VARE** framework with $\mathcal{L}\_{VARE}$ instead of cross entropy loss, and simplified the computation of $F\_{i}$ as $F\_{i} = \mathbb{I}( \mathcal{L}\_{VARE} > \gamma)$, where $\gamma$ is the loss value of the bottom 25%. The experimental results, shown in the table below, demonstrate that our method could also effectively erases the target concept on other visual autoregressive model.
>
> ||ACC$_{e} \downarrow$||ACC$_{i}\uparrow$||
> |-|-|-|-|-|
> ||Original|Ours|Original|Ours|
> |CassettePlayer|99.4|5.2|91.2|91.0|
> |ChainSaw|99.0|2.0|91.2|90.8|
> |Church|98.0|6.5|91.3|91.0|
> |GasPump|97.4|3.4|91.4|89.3|
> |Tench|78.8|0.8|93.5|92.7|
> |GarbageTruck|98.0|7.0|91.3|90.3|
> |EnglishSpringer|71.2|0.0|94.3|93.5|
> |GolfBall|100.0|1.8|91.1|90.7|
> |Parachute|100.0|1.0|91.1|90.2|
> |FrenchHorn|78.2|0.0|95.5|93.2|
>
>
>
> >**Question 3: The efficacy of the proposed erasure loss on style erasure task.**
>
> In deploying $\mathcal{L}\_{FCE}$, we calculate the proportion of erroneous bits in each token based on the loss function value, and optimize only the tokens whose error rate exceeds a predefined threshold. **This process is influenced solely by whether the text prompt, due to containing the target concept, causes a difference in visual tokens**. Therefore, regardless of whether the concept is abstract or concrete, our method is effective.
>
> **In Figure 3**, we show the token changes when erasing the “Van Gogh” concept, where the bluer regions indicate smaller loss values. It can be observed that, when handling abstract concepts, the VAR model still exhibits a similar loss distribution pattern across all scales. In addition to the results demonstrated in **Table 4**, we performed an ablation study on $\mathcal{L}\_{FCE}$ for the task of erasing style concepts, with the experimental results presented in the table below. These results show that using $\mathcal{L}\_{FCE}$ not only achieves stronger erasure effects but also further preserves the model’s overall generative capability. More discussions on style erasure task could be found in **Appendix E.3**.
>
> ||ACC$\downarrow$|FID$\downarrow$|CLIP$\uparrow$|
> |-|-|-|-|
> |Baseline|18.4|34.1|30.5|
> |+ $\mathcal{L}\_{FCE}$|8.8|33.4|31.2|

---

> > ### Comment · Reviewer_TYDT · 2025-11-25
> >
> > Thank you for the helpful clarifications.
> >
> > However, I would like to raise two further concerns related to **W3** and **Q3**:
> >
> > 1. **On the Generalization of a Fixed Threshold Across Diverse Concepts (Related to Weakness 3)**
> >
> > You mention that the filtering mechanism relies on a fixed threshold applied to the proportion of token-level error. My concern is that **different concepts (e.g., "Van Gogh" vs. "nudity") may exhibit distinct loss/error distributions** due to their semantic scope, visual patterns, or contextual usage in prompts.
> >
> > - Is there any statistical evidence that the token error distributions are comparable across concept types?
> > - If not, does applying a fixed threshold risk under-filtering for subtle concepts and over-filtering for dominant or global concepts?
> > - Would adaptive thresholding (e.g., percentile-based or normalized per-concept) offer improved generalization?
> >
> > **2. On the Effectiveness of Filtered Cross-Entropy for Global Style Erasure (Related to Question 3)**
> >
> > My original question intended to probe the suitability of the filtered cross-entropy mechanism for **style erasure**, which often involves **global** visual patterns (e.g., "Van Gogh" or "cyberpunk").
> >
> > In such cases, it is plausible that **most or even all tokens** may exceed the predefined error threshold due to the style being distributed throughout the image. If that happens:
> >
> > - Would the filtered cross-entropy loss essentially reduce to standard cross-entropy, thereby losing its intended selectivity?
> > - If so, how does the method avoid overfitting or loss of generality when erasing such globally entangled concepts?

---

> > > ### Author Response · Authors · 2025-11-25
> > >
> > > Thank you for your suggestion. In our response to **Question 3**, we have added ablation experiments comparing the use of $\mathcal{L}\_{FCE}$ and $\mathcal{L}\_{VARE}$ when erasing the **"Van Gogh" style concept**. Regarding the effect of the fixed threshold $\alpha$, we need to clarify that our filtering mechanism **does not select a fixed proportion of tokens** based on their **token-level** loss function values. Instead, our method filters tokens by using the **bit-level** binary cross-entropy loss value as an implicit semantic indicator. **Only tokens with a bit error rate exceeding $\alpha$ are selected for optimization**, as described in lines 249-252. Therefore, **the proportion of tokens filtered for optimization differs across concepts, as our proposed method is an implicitly dynamic filtering mechanism**.
> > >
> > > To support this, we provide a **fixed token-level filtering mechanism** reference baseline, where we fix the non-optimized tokens with loss values in the bottom 25% and present two ablation experiments on **NSFW** and **Style** concept erasure tasks. As shown in the tables below, our method demonstrates effective generalization across all concepts compared to fixed token-level filter mechanism.
> > >
> > > |||NSFW Erasure||
> > > |-|-|-|-|
> > > ||ASR$\downarrow$|FID$\downarrow$|CLIP$\uparrow$|
> > > |$\mathcal{L}\_{CE}$|**7.2**|33.6|30.9|
> > > |$with$ token-level filter|7.2|33.3|30.9|
> > > |$with$ bit-level filter|7.4|**32.8**|**31.3**|
> > >
> > > |||Style Erasure||
> > > |-|-|-|-|
> > > ||ACC$\downarrow$|FID$\downarrow$|CLIP$\uparrow$|
> > > |$\mathcal{L}\_{CE}$|10.6|34.2|30.8|
> > > |$with$ token-level filter|10.2|33.9|31.2|
> > > |$with$ bit-level filter|**8.8**|**33.4**|**31.2**|
> > >
> > > We agree with your insight into the style erasure task, during the early stage of optimization, the number of tokens to be optimized is indeed distributed across the global image and occupies a large proportion. **We optimize all tokens with a bit error rate exceeding $\alpha$**. In the later stages of optimization, as shown in **Figure 3**, the number of erroneous bits in low-frequency subject tokens at each scale sharply decreases. **Our filtering mechanism allows the optimization process to focus only on high-frequency style tokens that still contain many erroneous bits**, enabling dynamic filtering during the optimization process.
> > >
> > > The selection of our threshold $\alpha$ is based on the self-correction feature introduced during the pre-training phase of Infinity, a characteristic acquired through pretraining on large-scale datasets containing multiple concepts, which effectively generalizes to different concept erasure tasks, and we also provide related ablation experiments on $\alpha$ in **Table 5**.

---

> > > > ### Comment · Reviewer_TYDT · 2025-11-26
> > > >
> > > > Thank you for your clarifications! Now I fully understand your solution. I decide to maintain my positive score.

---

> > > > > ### Author Response · Authors · 2025-11-26
> > > > >
> > > > > We are very delighted to see that your concerns have been addressed. We appreciate the support. Your feedback is very helpful for us to improve our work, and we will incorporate the additional discussion in the revision based on your advice. If there are any further questions, don't hesitate to post here for discussion. Thank you.

---

### Official Review · Reviewer_yGxY · 2025-10-30

**Soundness:** 2
**Presentation:** 1
**Contribution:** 2
**Rating:** 2
**Confidence:** 3

**Summary:**

This paper presents the VARE framework and its variant, S-VARE, for concept erasure in visual autoregressive models. The approach stabilizes training and achieves concept erasure through a filtered cross-entropy loss.

**Strengths:**

1. This paper introduces concept erasure into visual autoregressive models, employing auxiliary visual tokens for stable training, filtered cross-entropy for precise erasure, and retention loss to preserve image quality.

2.The proposed method demonstrates effectiveness for sensitive concepts/objects/styles, with sufficient experimental support and clear comparisons against baselines, presenting a comprehensive analysis.

**Weaknesses:**

Major Weakness

1.In line 59, the differential prompt method should be supported with an appropriate citation. In addition, it is recommended to include comparative experiments to demonstrate the superiority of the proposed VARE framework, which leverages auxiliary target tokens, over the differential prompt method, supported by quantitative results.

2.In lines 211–215, the statement "This fundamental mismatch leads to degrading the fidelity of the main subject" lacks citation or experimental evidence. Moreover, the claim that " it is not directly applicable to autoregressive models" is not reasonable, as Equation 7 is an MSE-based loss and can also be applied to VAR models. Comparative experiments between the VARE framework trained with Equation 7 and the S-VARE method should be included to demonstrate the superiority of S-VARE.

3.Please further clarify how Equation 7 improves upon Equation 6. Specifically, describe the internal structure of r^{ori} _{<i}, including how tokens interact with the VAR model.

4.The relationship between the cross-attention mentioned in line 203 and c^{*} should be explicitly illustrated in Figure 1.

5.Line 234 mentions the "characteristics of infinity", followed by a proposed solution, but the concept ("characteristics of infinity") is not clearly defined or explained in the text. Relevant descriptions should be provided.

6.Please describe the novel modifications (structure) in the VARE framework within the Methods section. For example, explain how the auxiliary target tokens are embedded into the VAR model and provide the corresponding equations.

Minor weakness

1.In lines 195–202, the symbols in the main text are inconsistent with those in Equation (7) and Figure 1 and need to be corrected. For example, r_{ori}^{*} and r_{ori} mentioned in line 167 do not appear in the equation, and their meaning is unclear. In line 202, the text refers to "see left of Figure 1", but r^{ori} is not labeled in Figure 1, and the symbol r is not explained.

2.What does the subscript in r_{<I}​ represent in Equation 6? Specifically, what is the meaning of r_{<I}?

3.Figure 1 presents the overall framework, the text in line 155 should be revised to "the overall framework is shown in Figure 1".

4.Based on the text, "VARE" refers to a framework and "S-VARE" to a method. The phrase "erasure method VARE" in lines 78–80 should be corrected to "erasure method S-VARE".

5.In Figure 1, replace -log1/2 with γ.

**Questions:**

1.In line 188, the statement "optimization with Eq. (6) introduces errors as illustrated in the left column of Figure 2" means that the VAR model trained with Equation 6 fails to generate images properly, as shown in the left column of Figure 2. Furthermore, please provide a detailed description of the diffusion-based CE method, including how text is embedded into the VAR framework and whether a pre-trained model is used. Additionally, clarify in the main text the specific implementations and differences between the methods shown in Figure 2.

2.How does Figure 3 support the conclusion that "VAR maintains consistent optimization objectives across scales"? Please further clarify the content presented in the figure.

---

> ### Author Response · Authors · 2025-11-19
>
> We would like to express our gratitude for your valuable suggestions and insightful feedback of our paper. Our point-by-point responses are provided below.
>
> >**Weakness 1: The fundamental differential prompt setting in the field of concept erasure.**
>
> Differential prompts refer to prompt $c^{*}$ that contain the target concept and prompt $c$ that do not. **Fine-tuning a model using differential prompts is a standard practice in the field of concept erasure**. We provide the differential prompt training dataset used in **Appendix D** and will include citations to existing concept erasure works in line 59.
>
> Since all concept erasure works, including ours, use differential prompt pair $c^{*}$ and $c$ as text inputs, the novelty of our work lies in the concept erasure framework specifically designed for VAR models, along with the introduction of new loss functions. A quantitative and qualitative comparison with existing concept erasure methods is shown in **Table 1** and **Figure 4**.
>
>
>
> >**Weakness 2: The clarification of the contained ablation study.**
>
> First, we explain in lines 214-215 that the MSE used in Equation 7 is not suitable for Infinity because of the BSQ employed by Infinity, which makes its prediction targets reside in a **probability space**. Additionally, in line 458 of Section 5.3, we discuss the ablation experiments using the native **VARE** version of Equation 7, the erasure loss function $\mathcal{L}\_{FCE}$ in **S-VARE** (Equation 10), and the preservation loss function $\mathcal{L}\_{Pre}$ (Equation 11), as shown in **Table 4** and **Figure 6**. The quantitative experimental results validate the authenticity of the issues discussed and demonstrate the further superiority of **S-VARE** within the **VARE** framework proposed by us.
>
>
>
>
>
> >**Weakness 3: The clarification of the improvement of $\mathcal{L}\_{VARE}$.**
>
> In lines 185-189, we explain that the generation process of VAR heavily relies on the previous layer's $r\_{<i}$. Therefore, changes to target tokens during the concept erasure optimization process can introduce cumulative errors in the generation of subsequent scale tokens. To address this, we use the tokens $r^{ori}$ generated by the pre-trained model with $c$ as the text input as the input tokens for each scale of the fine-tuned model, and jointly predict the next scale token with $c^{*}$.
>
> As shown in the left part of **Figure 1**, we use $r\_{i}$ as the visual token input $f\_{i}$ of the fine-tuned model instead of reproducing the visual token. The improvement of Equation 7 over Equation 6 in terms of visual performance is shown in the comparison between the rightmost and leftmost columns of **Figure 2**. Without the additional visual tokens, the model completely loses its generation capability, thus demonstrating the effectiveness of our approach.
>
>
>
> >**Weakness 4: The relationship between the cross-attention and the text input.**
>
> Thank you for your suggestion. The cross-attention in line 203 refers to the cross-attention module of the visual transformer, which computes the cross-attention between visual tokens $r\_{i}$ and the text tokens generated by $c^{*}$. We did not modify the cross-attention module and will include this clarification in the final version.
>
>
>
>
> >**Weakness 5: The clarification of the contained description.**
>
> In lines 235 and 143-150, we provide a detailed explanation of how Infinity deploys BSQ on top of the VAR model, addressing the issue of a large codebook through quantization, thereby enabling large-scale text-to-image generation. The characteristic of Infinity referenced in line 234 is its use of BSQ, which transforms the training objective into a classification problem at the bit scale. The effectiveness of the $\mathcal{L}\_{FCE}$ designed based on this characteristic is also validated in **Table 4**.
>
> >**Weakness 6: The novel modification of the proposed method.**
>
> In **Section 4.1**, we explain through **Figure 2** and the accompanying description that, without additional visual tokens $r^{ori}$, the VAR model suffers from cumulative errors, causing the generation process to collapse. Specifically, before introducing $r^{ori}$, the fine-tuned model uses the visual tokens generated by the previous scale $r\_{<i}$ as input to predict the next scale visual tokens $r\_{i}$. Our **VARE** framework replaces the visual tokens $r\_{i}$ at each scale of the fine-tuned model with the visual tokens $r\_{i}^{ori}$ generated by the pretrained model using $c$ as the text input. This provides a foundational anchor for the optimization process, limiting it to considering only the interaction between the visual tokens and the text tokens generated by $c^{*}$.

---

> ### Author Response · Authors · 2025-11-19
>
> >**Weakness 7 & Weakness 8: The clarification of the contained symbol definition.**
>
> In line 131, we define $r = \lbrace r\_{1}, r\_{2}, \cdots, r\_{K} \rbrace$ as the residual token map of all scales produced by the VAR model, while in line 138, we define $r\_{<k} =  \lbrace r\_{1}, r\_{2}, \cdots, r\_{k-1} \rbrace$ as the set of residual token maps from all scales prior to the current scale $k$. Therefore, in line 197, the $r^{ori, * } $ and $r^{ori}$ mentioned refer to the full set of visual tokens generated by the pre-trained model using $c^{*}$ and $c$ as text inputs, respectively. In Equation 7, the VAR model predicts the next scale's token based on all tokens from previous scales, which is why $r\_{<i}$ is used instead of $r$, consistent with our definition in **Section 3**.
>
> Furthermore, the middle column of **Figure 2** shows that when $r^{ori,*}$ is used as the additional visual token, the model still fails to generate images correctly. This explains why $r^{ori, *}$ does not appear in Equation 7. In **Figure 1**, $r\_{i}$ corresponds to $r\_{i}^{ori}$. We will include this clarification in the final version.
>
>
>
> >**Weakness 9 & Weakness 10 & Weakness 11: Correction of the typo error.**
>
> Thanks for your suggestion. We will correct these typo error in the final version.
>
>
>
>
>
> >**Question 1: The clarification of the setting of diffusion-based method.**
>
> In line 187, we point out that the results obtained by optimizing VAR using Equation 6 are shown in the leftmost column of **Figure 2**. Without using our proposed **VARE** framework, all diffusion model-based concept erasure methods lead to a collapse in the model's generation capability. Additionally, in **Appendix A**, we provide the implementations and corresponding formulas for all concept erasure methods used as baselines.
>
> For all concept erasure methods, the text is only used as the text prompt input required for the text-to-image task in the VAR model, and there is no difference between methods in this regard. The goal of concept erasure in this field is to efficiently erase specific concepts in the pre-trained model, making the pre-trained model necessary for all methods. **These two experimental setups are standard practices in the concept erasure field, which we also adhere to**.
>
>
>
>
>
> >**Question 2: The detailed analysis of the visualization results.**
>
> In **Figure 3**, we present the loss values corresponding to each token at different scales and optimization iterations. It can be observed that, despite the initial differences in the loss distributions of tokens at different scales, after optimization with the concept erasure loss function, all scales exhibit the same loss distribution pattern. This is reflected in the second row, where the blue portions are concentrated around the middle area. Therefore, we propose using $\mathcal{L}\_{FCE}$ for more precise optimization with restrictions. The relevant experimental results are shown in **Table 4** and **Figure 6**. We will include the corresponding clarifications in the final version.

---

> ### Author Response · Authors · 2025-11-27
>
> Dear Reviewer,
>
> I hope this message finds you well. As the discussion period is nearing its end with less than five days remaining, I wanted to ensure we have addressed all your concerns satisfactorily. If there are any additional points or feedback you'd like us to consider, please let us know. Your insights are invaluable to us, and we're eager to address any remaining issues to improve our work.
>
> Thank you for your time and effort in reviewing our paper.

---

### Official Review · Reviewer_dr9A · 2025-11-01

**Soundness:** 3
**Presentation:** 3
**Contribution:** 3
**Rating:** 6
**Confidence:** 3

**Summary:**

This manuscript addresses a critical gap in the safety of Visual Autoregressive (VAR) models for text-to-image generation: existing Concept Erasure (CE) techniques, primarily designed for diffusion models, fail to generalize to VARs due to their next-scale token prediction paradigm. VAR models, which offer advantages in image quality and generation speed, remain vulnerable to generating sensitive content (NSFW, copyright-sensitive material) post-training, and retraining for new undesirable concepts is computationally prohibitive.
To solve this, the authors propose two core solutions: (1) VARE (VAR Erasure framework), which leverages auxiliary visual tokens to reduce fine-tuning intensity and mitigate cumulative errors across scales— a key limitation of directly adapting diffusion-based CE; (2) S-VARE, an enhanced method built on VARE that incorporates a filtered cross-entropy loss (L_FCE) (to precisely identify and minimally adjust unsafe tokens) and a preservation loss (L_Pre) (to maintain semantic fidelity and avoid language drift/diversity reduction). Extensive experiments show the approach erases 97% of sensitive concepts with <2% CLIP score degradation, closing the safety gap for VAR models.

**Strengths:**

1.	Targeted Solution to a Framework-Specific Gap: The manuscript clearly identifies why diffusion-based CE methods fail for VARs and designs VARE/S-VARE to align with VAR’s intrinsic dynamics. The use of auxiliary visual tokens to counter scale-wise error accumulation directly addresses a core technical barrier.
2.	Loss Function Design Aligns with VAR Characteristics: The filtered cross-entropy loss (L_FCE) is tailored to VAR’s binary spherical quantization (BSQ) by measuring bit-wise discrepancies, avoiding the instability of diffusion-style MSE loss on discrete token spaces. The preservation loss (L_Pre), which uses KL divergence to align pre-trained and fine-tuned model outputs, effectively mitigates common fine-tuning side effects.
3.	Comprehensive Experimental Coverage: The authors test S-VARE across three critical CE tasks and validate robustness using adversarial datasets and real-user prompts —a rare focus on real-world deployment risks. Quantitative results and qualitative visualizations (Figure 4) provide clear evidence of the method’s effectiveness.

**Weaknesses:**

1.	Ambiguity in Auxiliary Token Selection Logic: The manuscript states VARE uses tokens from prompts c (neutral) and c⁎ (with target concept) as auxiliary inputs but provides no details on how to handle scenarios where c and c⁎ have large semantic gaps (e.g., "a violent car crash" vs. "a peaceful park").
2.	Limited Generalization Across VAR Models and Data: The authors only use Infinity-2B as the base VAR model, with no validation on other VAR variants (e.g., models with different scale prediction strategies or non-BSQ quantization).

**Questions:**

1.	Regarding Weakness 1: For semantically distant c/c⁎ pairs (e.g., "a violent car crash" vs. "a peaceful park"). How do auxiliary visual tokens avoid misaligning non-target concepts (e.g., cannot distinguish target "violent" and non-target "car crash" )? Is there quantitative data on the model’s ability to distinguish target vs. non-target tokens in such cases?
2.	Addressing Weakness 2: Have you tested S-VARE on VAR models with non-BSQ quantization (e.g., traditional VQ-VAE)?
3.	The manuscript sets the filter ratio α in L_FCE to 25% based on Infinity’s self-correction range. For VAR models with different self-correction thresholds (e.g., 10% or 40%), what adaptive strategy would you recommend for α?

---

> ### Author Response · Authors · 2025-11-19
>
> Thank you for dedicating your time and effort to reviewing our paper. We are deeply encouraged by your positive comments and recognition of our work. Below, we provide detailed responses to address your concerns.
>
> >**Weakness 1 & Question 1: The experimental setting of the differential prompt pair.**
> s.
> Thank you for your feedback. The training prompt pairs $c^{*}$ and $c$ used to erase specific target concepts are constructed based on existing workflows [1] [2], utilizing LLMs for keyword substitution while ensuring semantic consistency. As a result, there are no significant semantic discrepancies in the prompt pairs during the training process. The related design details and visualizations of the training dataset are provided in **Appendix D** and **Figure 9**.
>
>
>
>
> >**Weakness 2 & Question 2: Experimental results on other visual autoregressive model.**
>
> Our work addresses the issue of sensitive concepts generated by text-to-image autoregressive models, specifically Infinity. As Infinity is currently the only model supporting large-scale text-to-image tasks, we validated our method on Infinity-2B. Following your suggestion, we also conducted concept erasure experiments on the larger Infinity-8B model using the same task setup as in **Table 1**. The results, shown in the table below, demonstrate that our method effectively erases the target concept while preserving the model’s overall generation ability.
>
> |||NSFW Erasure|||
> |-|-|-|-|-|
> ||Sen.$\downarrow$|Com.$\uparrow$|FID$\downarrow$|CLIP$\uparrow$|
> |Original|167|135|25.3|33.6|
> |Ours|6|84|27.6|33.2|
>
> |||Object Erasure|||
> |-|-|-|-|-|
> ||ACC$_{e}\downarrow$|ACC$_{i}\uparrow$|FID$\downarrow$|CLIP$\uparrow$|
> |Original|95.1|80.2|25.3|33.6|
> |Ours|6.5|80.0|26.8|33.4|
>
> |||Style Erasure||
> |-|-|-|-|
> ||ACC$\downarrow$|FID$\downarrow$|CLIP$\uparrow$|
> |Original|78.0|25.3|33.6|
> |Ours|6.6|26.7|33.5|
>
> We also conducted experiments on the original VAR-d30 model, which differs from Infinity by using VQ-VAE and supporting label-based generation on the ImageNet-1K dataset, rather than BSQ for text-to-image tasks. For the object concept erasure, we implemented the VARE framework with $\mathcal{L}\_{VARE}$ instead of cross-entropy loss, simplifying the computation of $F\_{i}$ as $F\_{i} = \mathbb{I}( \mathcal{L}\_{VARE} > \gamma$), where $\gamma$ is the bottom 25% of the loss function value. The results, shown in the table below, demonstrate that our method effectively erases the target concept.
>
> ||ACC$_{e} \downarrow$||ACC$_{i}\uparrow$||
> |-|-|-|-|-|
> ||Original|Ours|Original|Ours|
> |CassettePlayer|99.4|5.2|91.2|91.0|
> |ChainSaw|99.0|2.0|91.2|90.8|
> |Church|98.0|6.5|91.3|91.0|
> |GasPump|97.4|3.4|91.4|89.3|
> |Tench|78.8|0.8|93.5|92.7|
> |GarbageTruck|98.0|7.0|91.3|90.3|
> |EnglishSpringer|71.2|0.0|94.3|93.5|
> |GolfBall|100.0|1.8|91.1|90.7|
> |Parachute|100.0|1.0|91.1|90.2|
> |FrenchHorn|78.2|0.0|95.5|93.2|
>
>
>
> >**Question 3: The choice of the filter ratio $\alpha$.**
>
> Thank you for your feedback. Since Infinity introduces a random flipping range of 0% to 30% during training to enhance the model's self-correction capability, this is a model characteristic introduced during the pre-training phase. Thus we use a filter to limit the optimization strength in the actual deployment to avoid over-optimization. Ablation experiments on the filtering rate $\alpha$ are shown in **Table 5**.
>
> Based on the experimental results, we believe that when the model has stronger self-correction ability, the corresponding $\alpha$ value should be increased to minimize the impact on the model's overall generation capability. However, we also need to consider the ablation results to avoid selecting an excessively high $\alpha$ value, which may lead to ineffective fine-tuning. We look forward to further discussion on this issue with you.
>
>
>
>
> [1] EraseAnything: Enabling Concept Erasure in Rectified Flow Transformers. In ICML, 2025.
>
> [2] EAR: Erasing Concepts from Unified Autoregressive Models. In arXiv, 2025.

---

> ### Author Response · Authors · 2025-11-27
>
> Dear Reviewer,
>
> I hope this message finds you well. As the discussion period is nearing its end with less than five days remaining, I wanted to ensure we have addressed all your concerns satisfactorily. If there are any additional points or feedback you'd like us to consider, please let us know. Your insights are invaluable to us, and we're eager to address any remaining issues to improve our work.
>
> Thank you for your time and effort in reviewing our paper.

---

### Official Review · Reviewer_9kz9 · 2025-11-01

**Soundness:** 3
**Presentation:** 3
**Contribution:** 3
**Rating:** 6
**Confidence:** 3

**Summary:**

This paper identifies a safety gap in modern Visual Autoregressive (VAR) text-to-image models, noting that existing concept erasure (CE) methods designed for diffusion models fail when applied to VARs due to their different (next-scale token prediction) architecture. The authors propose VARE, a foundational framework that enables stable erasure by using auxiliary tokens, and S-VARE, a specific method built upon it. S-VARE introduces two novel components: a "filtered cross entropy loss" (LFCE) specifically designed for the quantization-based token prediction of VARs, and a "preservation loss" (LPre) to prevent the fine-tuning process from degrading unrelated concepts or reducing image diversity.

**Strengths:**

1. The introduction clearly diagnoses why diffusion-based CE methods fail—they are not designed for the "visual token" prediction paradigm of VARs and lead to "cumulative errors" and "severe degradation."

2. The proposed solution appears well-reasoned and specifically tailored to the VAR architecture. Instead of retrofitting a Mean Squared Error (MSE) loss (for noise), it proposes a cross-entropy-based loss (for tokens/probabilities), which seems far more appropriate for the described model.

3. The inclusion of a "preservation loss" (LPre) is a strong point. It shows the authors are not just focused on the primary goal (erasure) but also on the critical side effects of fine-tuning, such as "language drift" and loss of diversity, which are common failings of naive erasure methods.

4. The headline metrics (97% erasure with <2% degradation in CLIP score) are impressive and, if validated, would represent a significant achievement in closing the safety gap.

**Weaknesses:**

1. The justification for the LFCE loss is heavily tied to the "binary spherical quantization (BSQ)" used in the Infinity model. It's not clear how generalizable this method is to other or future VAR models that might not use this specific quantization scheme.

2. The paper discusses erasing "sensitive" or "undesirable" concepts. This is vague. The method's difficulty would vary greatly between erasing a simple object (e.g., a specific logo) versus a complex, abstract, or compositional concept (e.g., "a violent action").

**Questions:**

1. What was the range of concepts used for the erasure experiments? Did you test the method on simple objects, artistic styles, specific identities, and more abstract concepts?

2. How did you balance the weights of the filtered erasure loss (LFCE) and the preservation loss (LPre)? Is there a risk that the preservation loss could "protect" parts of an unwanted concept, making erasure more difficult?

---

> ### Author Response · Authors · 2025-11-19
>
> We sincerely express our gratitude for dedicating your valuable time to providing insightful suggestions that can enhance our paper. Your praise regarding our methodology, experiments, and contributions have greatly encouraged and motivated us. Our detailed responses to all of your concerns are presented below.
>
> >**Weakness 1: Experimental results on other visual autoregressive model.**
>
> Thank you for your feedback. Since Infinity is currently the only visual autoregressive model that supports large-scale text-to-image tasks, our method aims to address the issue of deploying existing diffusion model-based methods effectively on VAR models, which face challenges due to architectural differences. **This requires considering model-specific characteristics, such as BSQ, and making specific design choices**. The concept erasure works [1] [2] that has transitioned from Stable Diffusion to FLUX also validates our design approach. Following your suggestion, we conducted object concept erasure experiments on the original **VAR-d30** model, which does not use BSQ and only generates ImageNet-1K images.
>
> Since VAR-d30 does not use BSQ, we implemented the VARE framework with $\mathcal{L}\_{VARE}$ instead of cross-entropy loss, simplifying the computation of $F{i}$ as $F\_{i} = \mathbb{I}( \mathcal{L}\_{VARE} > \gamma)$, where $\gamma$ is the bottom 25% of the loss function value. The experimental results, as shown in the table below, demonstrate that our method also accurately erases the target concept.
>
> ||ACC$_{e} \downarrow$||ACC$_{i}\uparrow$||
> |-|-|-|-|-|
> ||Original|Ours|Original|Ours|
> |CassettePlayer|99.4|5.2|91.2|91.0|
> |ChainSaw|99.0|2.0|91.2|90.8|
> |Church|98.0|6.5|91.3|91.0|
> |GasPump|97.4|3.4|91.4|89.3|
> |Tench|78.8|0.8|93.5|92.7|
> |GarbageTruck|98.0|7.0|91.3|90.3|
> |EnglishSpringer|71.2|0.0|94.3|93.5|
> |GolfBall|100.0|1.8|91.1|90.7|
> |Parachute|100.0|1.0|91.1|90.2|
> |FrenchHorn|78.2|0.0|95.5|93.2|
>
>
>
> >**Weakness 2 & Question 1: The range of concepts used for validate our method.**
>
> To validate the effectiveness of our proposed concept erasure method, we have designed experiments on **NSFW**, **object**, and **style** concepts, with both quantitative and qualitative results presented in **Table 1** and **Figure 4**.
>
> - For NSFW concepts, we conducted additional experiments on multiple real-user and adversarial datasets, including I2P, MMA, and R-A-B.
>
> - For object concepts, we used ten object categories from ImageNette as target concepts, as shown in **Table 2**.
>
> - For style concepts, we selected artistic styles supported by Infinity, such as Van Gogh, Monet, etc., as shown in **Table 1** and **Appendix E.3**.
>
> These settings align with the standard evaluation protocols in the field of concept erasure [3] [4]. Our method achieves good results for both abstract and entity concepts. Following your suggestion, we used the experimental setup from TRCE [5] and applied I2P to erase three abstract concepts, including **violence**, **self-harm**, and **shocking**. The experimental results as shown in the table below demonstrate that our method still achieves effective erasure.
>
> ||Violence|Self-harm|Shocking|
> |-|-|-|-|
> |Original|235|176|313|
> |Ours|13|9|17|
>
> >**Question 2: How to balance the erasure and preservation loss.**
>
> In the computation of $\mathcal{L}\_{FCE}$ and $\mathcal{L}\_{Pre}$, we use $c^{*}$ and $c$ as text inputs, respectively, and align the outputs of the pretrained model using different loss function calculation methods, which leads to no inherent contradiction. The related ablation experiments are shown in **Table 4** and **Appendix C.3**.
>
> Additionally, we provide in the table below the impact of using different $\beta$ values in the overall loss function $\mathcal{L}\_{FCE} + \beta \mathcal{L}\_{Pre}$. It can be observed that, while using $\mathcal{L}\_{Pre}$ on top of $\mathcal{L}\_{FCE}$ almost does not affect the erasure performance, it effectively improves the model's performance. In contrast, **a small $\beta$ value results in ineffective alignment of the pre-trained model’s output distribution, while a large $\beta$ leads to insufficient optimization of the $\mathcal{L}\_{FCE}$**. Therefore, we simply choose $\beta=1$ as our standard experimental setting.
>
> ||ACC$_{e}\downarrow$|ACC$_{i}\uparrow$|FID$\downarrow$|CLIP$\uparrow$|
> |-|-|-|-|-|
> |$\beta=0.01$|4.1|74.3|32.2|31.2|
> |$\beta=0.1$|4.5|75.3|31.5|31.4|
> |**$\beta=1$**|4.4|75.7|31.5|31.6|
> |$\beta=5$|5.0|75.5|31.5|31.6|
> |$\beta=10$|5.2|75.8|31.4|31.6|
>
> [1] EraseAnything: Enabling Concept Erasure in Rectified Flow Transformers. In ICML, 2025.
>
> [2] Minimalist Concept Erasure in Generative Models. In ICML, 2025.
>
> [3] Erasing Concepts from Diffusion Models. In ICCV, 2023.
>
> [4] One Image is Worth a Thousand Words: A Usability Preservable Text-Image Collaborative Erasing Framework. In ICML, 2025.
>
> [5] Trce: Towards reliable malicious concept erasure in text-to-image diffusion models. In ICCV, 2025.

---

> ### Author Response · Authors · 2025-11-27
>
> Dear Reviewer,
>
> I hope this message finds you well. As the discussion period is nearing its end with less than five days remaining, I wanted to ensure we have addressed all your concerns satisfactorily. If there are any additional points or feedback you'd like us to consider, please let us know. Your insights are invaluable to us, and we're eager to address any remaining issues to improve our work.
>
> Thank you for your time and effort in reviewing our paper.

---

### Official Review · Reviewer_ZPfw · 2025-11-03

**Soundness:** 3
**Presentation:** 3
**Contribution:** 3
**Rating:** 6
**Confidence:** 5

**Summary:**

This paper proposes VARE and S-VARE, the first frameworks for concept erasure in VAR text-to-image models. VARE introduces auxiliary visual tokens to stabilize optimization and mitigate cumulative errors. S-VARE integrates a filtered cross-entropy loss and a preservation loss to achieve surgical concept removal while maintaining generation quality. Experiments show that this method has certain effectiveness.

**Strengths:**

1. This paper focuses on the problem of concept erasure in  VAR text-to-image models, addressing a significant gap in current research.
2. Experiments cover various erasure tasks and multiple datasets, with visualizations further demonstrating the effectiveness of the proposed method.

**Weaknesses:**

1. All experiments in this paper are conducted based on Infinity-2B. It lacks validation on other autoregressive architectures, which may limit the generalization of the results.
2. The authors aim to use $L_{FCE}$ to promote erasure and $L_{Pre}$ to preserve the model’s alignment with the original distribution. However, these two objectives are inherently conflicting, which may lead to gradient interference. The authors should clarify how the balance between them is achieved.
3. It remains unclear whether the proposed method can preserve the invariance of irrelevant factors in the original image during the erasure process. For example, after erasing nudity, can the method ensure that the person’s identity in the image remains unchanged?
4. The competitors are mostly from 2024. To my knowledge, several new SOTA methods have emerged in 2025. It is recommended that the authors include comparisons with them. If such methods cannot be adapted to autoregressive architectures, this limitation should be explicitly discussed in the Related Work.

**Questions:**

In addition to the issues mentioned in the Weaknesses, I have a few more concerns:

1. What are the time and space complexities of the proposed method?
2. Does erasing multiple concepts lead to cumulative negative effects, such as a decline in generation quality as the number of erased concepts increases? While I understand that there is no free lunch and that concept erasure will inevitably affect the original generation quality, I still hope the authors can provide a quantitative evaluation of this trade-off.

---

> ### Author Response · Authors · 2025-11-19
>
> We sincerely thank you for the precious time and effort in providing a wealth of suggestions to enhance the quality of our paper. We have carefully read all the comments and provide detailed point-by-point responses as follows. Hopefully, we can adequately address your concerns.
>
> >**Weakness 1: Experimental results on other visual autoregressive models.**
>
> Thank you for your suggestion. Our work focuses on addressing the issue of sensitive concepts generated by text-to-image visual autoregressive models, i.e., Infinity. Since Infinity is currently the only model that supports large-scale text-to-image tasks, we validated our proposed method on Infinity-2B. Based on your suggestion, we also conducted concept erasure experiments on a larger model, **Infinity-8B**, using the same task setup as in **Table 1**. As shown in the table below, our method still effectively erases the target concept while preserving the model's overall generation ability.
>
> |||NSFW Erasure|||
> |-|-|-|-|-|
> ||Sen.$\downarrow$|Com.$\uparrow$|FID$\downarrow$|CLIP$\uparrow$|
> |Original|167|135|25.3|33.6|
> |Ours|6|84|27.6|33.2|
>
> |||Object Erasure|||
> |-|-|-|-|-|
> ||ACC$_{e}\downarrow$|ACC$_{i}\uparrow$|FID$\downarrow$|CLIP$\uparrow$|
> |Original|95.1|80.2|25.3|33.6|
> |Ours|6.5|80.0|26.8|33.4|
>
> |||Style Erasure||
> |-|-|-|-|
> ||ACC$\downarrow$|FID$\downarrow$|CLIP$\uparrow$|
> |Original|78.0|25.3|33.6|
> |Ours|6.6|26.7|33.5|
>
>
> Additionally, we conducted further experiments on the original **VAR-d30** model. Unlike Infinity, which uses BSQ for text-to-image tasks, VAR-d30 uses only VQ-VAE and supports label-based generation on the ImageNet-1K dataset. Therefore, we conduct an experiment under the setting of erasing the object concept. Since VAR-d30 does not use BSQ, we directly implemented **VARE** framework with the $\mathcal{L}\_{VARE}$ instead of cross entropy loss, and simplified the computation of $F\_{i}$ as $F\_{i} = \mathbb{I}( \mathcal{L}\_{VARE} > \gamma$), where $\gamma$ is set as the loss function value of the bottom 25%. The experimental results, as shown in the table below, demonstrate that our method also accurately erases the target concept on other visual autoregressive model.
>
> ||ACC$_{e} \downarrow$||ACC$_{i}\uparrow$||
> |-|-|-|-|-|
> ||Original|Ours|Original|Ours|
> |CassettePlayer|99.4|5.2|91.2|91.0|
> |ChainSaw|99.0|2.0|91.2|90.8|
> |Church|98.0|6.5|91.3|91.0|
> |GasPump|97.4|3.4|91.4|89.3|
> |Tench|78.8|0.8|93.5|92.7|
> |GarbageTruck|98.0|7.0|91.3|90.3|
> |EnglishSpringer|71.2|0.0|94.3|93.5|
> |GolfBall|100.0|1.8|91.1|90.7|
> |Parachute|100.0|1.0|91.1|90.2|
> |FrenchHorn|78.2|0.0|95.5|93.2|
>
>
>
> >**Weakness 2: Clarification on the erasure and preservation loss**
>
> It is worth noting that, when computing $\mathcal{L}\_{FCE}$, we use a **prompt $c^{*}$ that contains the target concept to be erased** as the text input to the fine-tuned model, and align the output of the visual transformer with that of the pre-trained model when it takes as input a **prompt $c$ that does not contain the target concept**. In contrast, when computing $\mathcal{L}\_{Pre}$, we use the **same prompt $c$ without the target concept** as the text input to both the pre-trained and fine-tuned models and align their corresponding outputs. Since the **text inputs to the fine-tuned model used for computing the two loss terms are different**, no inherent contradiction arises. Furthermore, **Table 4** presents an ablation study on these two losses, showing that adding $\mathcal{L}\_{Pre}$ on top of $\mathcal{L}\_{FCE}$ can effectively improve the overall image generation quality while almost not affecting the erasure performance.
>
> >**Weakness 3: The preservation ability of the proposed method.**
>
> To precisely erase the target concept, we propose several solutions as follow.
>
> - When constructing the differential prompt pairs required for training, we modify only the target concept. The corresponding automatic construction strategy and visualizations of the generated images are provided in **Appendix D** and **Figure 9**, ensuring no significant alignment discrepancies.
>
> - We constrain the alignment process according to the value of the erasure loss when compute the$\mathcal{L}\_{FCE}$, for tokens that no longer require further modification, we stop updating them by using $F\_{i}$. This alleviates the image distortion caused by over-optimization during training, as demonstrated by the ablation results in **Table 5**.
>
> - We employ $\mathcal{L}\_{Pre}$ as a preservation loss to further align the output distributions of the fine-tuned and pretrained models, thereby mitigating the distribution shift that arises when aligning differential prompts during training. The corresponding ablation studies and visualizations are presented in **Table 4** and **Figure 6**.
>
> Additionally, the overall qualitative results are shown in **Figure 4**, where, compared with the baselines, only our method achieves the most precise erasure of the target concept while avoiding substantial changes to the generated images.

---

> ### Author Response · Authors · 2025-11-19
>
> >**Weakness 4: The choice of selected baseline methods.**
>
> Thank you for your feedback. **We are the first to identify the issues with VAR concept erasure and propose feasible solutions as the foundation for future work**. Therefore, the baselines we use are existing diffusion-based methods for different categories that have been transferred to VAR. For instance, FMN, AC, and ESD each propose optimization-based foundational loss functions, while UCE is a training-free foundational method. A more detailed discussion can be found in Appendix A.
>
> Subsequent concept erasure work builds on these methods by adding incremental optimization schemes. For example, RACE [1] incorporates adversarial training during the optimization process, and there have been numerous subsequent works [2] [3] that improve the adversarial training paradigm. AGE [4] dynamically selects the erased and retained concepts. These works do not modify the foundational loss function; therefore, **the techniques they propose are not meant to be compared with our foundational methods but rather serve as complementary approaches.**. The experimental results of applying these techniques on top of our method are presented in the table below, demonstrating that these enhancements further improve the performance of our proposed method, thereby proving the reliability and scalability of our approach as a foundational method.
>
> |||Sen.$\downarrow$|||ASR$\downarrow$||
> |-|-|-|-|-|-|-|
> ||Ours|Ours + RACE|Ours + AGE|Ours|Ours + RACE|Ours + AGE|
> |I2P|97|88|67|0.8|0.7|0.6|
> |MMA|101|84|52|3.5|2.8|1.4|
> |R-A-B|24|20|11|7.4|5.7|3.0|
> |Normal|5|5|0|3.0|3.0|0.0|
>
>
> >**Question 1: The training resources needed for the proposed method.**
>
> The training time required to erase a concept on a single Nvidia A6000 is 4 hours, with a memory usage of 38 GB.
>
>
>
> >**Question 2: The performance of multiple concepts erasure.**
>
> Thank you for your feedback. We have tested the effectiveness of multi-concept erasure on the task of erasing the object concept, and the experimental results are presented in **Table 7** of the **Appendix**. We demonstrate the effects of erasing different numbers of concepts. While erasing multiple concepts results in some reduction in erasure efficiency and generative capability compared to erasing a single concept, the impact is not substantial. This demonstrates the robustness of our method in multi-concept erasure tasks.
>
> [1] R.A.C.E.: Robust Adversarial Concept Erasure for Secure Text-to-Image Diffusion Model. In ECCV, 2024.
>
> [2] Erasing Undesirable Concepts in Diffusion Models with Adversarial Preservation. In NeurIPS, 2024.
>
> [3] STEREO: A Two-Stage Framework for Adversarially Robust Concept Erasing from Text-to-Image Diffusion Models. In CVPR, 2025.
>
> [4] Fantastic Targets for Concept Erasure in Diffusion Models and Where To Find Them. In ICLR, 2025.

---

> > ### Comment · Reviewer_ZPfw · 2025-11-26
> >
> > Thank you for the detailed response. I carefully read the rebuttal together with the comments from other reviewers and their corresponding replies. I have a further question regarding Weakness 3.
> >
> > - You use ACC as the evaluation metric for erasure accuracy and FID and CLIP scores for generation quality. However, the ideal goal of erasure is to remove only the target object while keeping the rest of the content unchanged. The metrics used in the paper cannot capture this preservation ability. I am curious whether other metrics can provide a quantitative measurement. For example, would it be feasible to compute the CLIP similarity between the images before and after erasure?

---

> > > ### Author Response · Authors · 2025-11-26
> > >
> > > Thanks for your suggestion. The metric for evaluating preservation ability we used (e.g., ACC$\_{i}$ for Object Erasure) is the same as the one employed in existing works [1] [2] [3], ensuring consistency with prior evaluation setting. Following your suggestion, we additionally selected a recent benchmark [4] that includes in-prompt evaluation using CLIP scores. Specifically, this evaluation involves using prompts containing a specific concept and prompts where the concept is removed as text inputs of fine-tuned model, then calculating the CLIP score between the output images.
> > >
> > > The erasure results for **Nudity**, **Object**, and **Style** concepts are shown in the table below. It can be observed that our method achieves significant performance improvement due to the introduction of filtering cross entropy loss and preservation loss, we also provide visualizations of the erased images in **Figure 4-6** and **Figure 10-13** in the paper as qualitative experimental results. We will include this additional content in the final version.
> > >
> > >
> > > ||Nudity|Object|Style|
> > > |-|-|-|-|
> > > |UCE|25.5|27.3|28.9|
> > > |FMN|25.4|26.8|28.4|
> > > |ESD-x|24.6|25.2|29.0|
> > > |AC-N|28.2|28.5|29.5|
> > > |Ours|**29.3**|**30.2**|**31.0**|
> > >
> > >
> > >
> > > [1] Erasing Concepts from Diffusion Models. In ICCV, 2023.
> > >
> > > [3] One Image is Worth a Thousand Words: A Usability Preservable Text-Image Collaborative Erasing Framework. In ICML, 2025.
> > >
> > > [4] Six-CD: Benchmarking Concept Removals for Benign Text-to-image Diffusion Models. In CVPR, 2025

---

> > > > ### Comment · Reviewer_ZPfw · 2025-11-26
> > > >
> > > > Thank you for your detailed and thoughtful responses. They effectively address my concerns and clarify the key points. The inclusion of the new competitors and metrics makes the paper more convincing. I will maintain my positive assessment.

---

> > > > > ### Author Response · Authors · 2025-11-26
> > > > >
> > > > > Your support is very important to us. We appreciate all suggestions given, and will make sure all issues are addressed in the revision. If you have any further questions, feel free to discuss them with us and we are more than happy to solve them. Thank you.

---

### Comment · Area_Chair_YobB · 2025-11-22

Dear Reviewers,

Thank you for your time and effort in reviewing submissions for ICLR  2026. As we begin the author-reviewer discussion process, we kindly remind you to submit your responses to the author rebuttals by **December  2**.


Your engagement in this discussion phase is crucial to ensuring a fair and thorough evaluation of each submission.

**Action Required**


- Carefully consider the authors’ rebuttal and any additional evidence they provide.

- Update your review (if applicable) to reflect your revised perspective.

-  **Discuss with the authors if further details are required**


Your AC

---

### Meta-Review · Area_Chair_gcqY · 2026-01-14

**Summary:**

This paper targets concept erasure for visual autoregressive (VAR) text-to-image models. It proposes a VAR Erasure framework (VARE) that stabilizes optimization via auxiliary visual tokens, and a concrete method (S-VARE) that combines a filtered cross-entropy loss to localize/adjust unsafe visual tokens with a preservation loss to reduce language drift and quality/diversity degradation. Experiments evaluate erasing NSFW, object, and style concepts and report strong erasure with limited quality loss.

**Reviewer Concerns:**

Addressed:
- Generalization beyond a single model (Infinity-2B) was a central concern (e.g., ZPfw, dr9A, 9kz9). The authors added cross-architecture results including Infinity-8B and an experiment on VAR-d30 (non-BSQ / VQ-VAE style)
- Loss-term interaction / how erasure vs preservation are balanced (ZPfw, 9kz9) received a concrete clarification: the rebuttal explains the two losses are computed under different prompt inputs and provides ablations over the preservation weight β.
- Preservation measurement / “surgical” invariance: ZPfw asked for a metric beyond ACC/FID/CLIP that better reflects content preservation; the authors added a CLIP-based in-prompt evaluation benchmark and ZPfw explicitly stated the new metrics/competitors made the paper more convincing.
- TYDT’s concerns appear resolved in discussion; TYDT explicitly states they “fully understand” and maintain their positive score.

Outstanding / partially addressed:
- Strength of “preservation” evidence for true factor invariance (e.g., “same identity after nudity removal”) remains only partially addressed. The added CLIP-based evaluation helps, but it’s not a direct “before/after same-scene” invariance test.
- Presentation/clarity issues (raised strongly by yGxY): yGxY raised many requests about missing citations, unclear definitions/notation, and unclear figures/equations; I do not see evidence in the visible thread that yGxY re-engaged or that these were fully verified as fixed during discussion, so this is still something there.

**Reviewer Scores:**

- Reviewer ZPfw (initial: 6): likely stays 6
- Reviewer 9kz9 (initial: 6, no reply): likely stays 6, possibly to 7 if fully engaged
- Reviewer dr9A (initial: 6, no reply): likely stays 6
- Reviewer yGxY (initial: 2, no reply): likely increases a bit
- Reviewer TYDT (initial: 6): stays 6, explicitly confirmed after discussion.

---

### Decision · Program_Chairs · 2026-01-26

Accept (Poster)